# Cyclin B1-Cdk1 facilitates MAD1 release from the nuclear pore to ensure a robust spindle checkpoint

Mark Jackman*, Chiara Marcozzi*⬤, Martina Barbiero, Mercedes Pardo⬤, Lu Yu⬤, Adam L. Tyson⬤, Jyoti S. Choudhary⬤, and Jonathon Pines⬤

**How the cell rapidly and completely reorganizes its architecture when it divides is a problem that has fascinated researchers for almost 150 yr. We now know that the core regulatory machinery is highly conserved in eukaryotes, but how these multiple protein kinases, protein phosphatases, and ubiquitin ligases are coordinated in space and time to remodel the cell in a matter of minutes remains a major question. Cyclin B1-Cdk is the primary kinase that drives mitotic remodeling; here we show that it is targeted to the nuclear pore complex (NPC) by binding an acidic face of the kinetochore checkpoint protein, MAD1, where it coordinates NPC disassembly with kinetochore assembly. Localized cyclin B1-Cdk1 is needed for the proper release of MAD1 from the embrace of TPR at the nuclear pore so that it can be recruited to kinetochores before nuclear envelope breakdown to maintain genomic stability.**

## Introduction

The rapid and complete reorganization of a cell at mitosis is one of the most striking events in cell biology, but we are only just beginning to understand how it is achieved. To understand the remarkable coordination required to remodel the interphase cell into a mitotic cell that is specialized to separate the genome equally into two daughter cells, we must elucidate the mechanisms by which the mitotic regulators disassemble interphase structures and promote the assembly of the mitotic apparatus. The conservation of much of the machinery through evolution has allowed us to establish that coordinated efforts of multiple protein kinases and phosphatases are required to remodel the cell. Chief among these are the activation of Cyclin B1-Cdk1, the major mitotic kinase in almost all organisms studied to date, and the concomitant inhibition of its antagonistic PP2A-B55δ phosphatase (Castilho et al., 2009; Gharbi-Ayachi et al., 2010; Mochida et al., 2010). Together, these drive the cell to enter mitosis. As the level of cyclin B1-Cdk1 activity rises in the cell, it triggers different events at different times (Gavet and Pines, 2010). But how this is achieved, and how the disassembly of interphase structures contributes to the assembly of mitosis-specific structures, are still largely unknown.

Although cyclin B1-Cdk1 was identified as the major mitotic kinase in the 1980s (Arion et al., 1988; Dorée and Hunt, 2002; Dunphy and Newport, 1989; Labbe et al., 1988; Meijer et al., 1989; Minshull et al., 1989), and a plethora of crucial substrates have been identified since then (Nigg, 1995; Wieser and Pines, 2015), it is remarkable that we still do not understand how cyclin B1-Cdk1 recognizes its substrates. Our knowledge is limited to the

minimal consensus sequence recognized by Cdk1 (S/T-P, optimally in the context of basic residues; De Bondt et al., 1993; Jeffrey et al., 1995; Brown et al., 1999; Alexander et al., 2011), and evidence that its associated Cks subunit, which also binds to Cdk2, preferentially recognizes phospho-threonines in a (F/I/L/P/V/W/Y-X-pT-P) consensus (McGrath et al., 2013). By contrast, we know that the major interphase cyclin-Cdk complexes, cyclins A and E, recognize many substrates through the Cy motif (RxL), which binds to the "hydrophobic patch" on the first cyclin fold (Schulman et al., 1998; Brown et al., 1999, 2007), and the D-type cyclins have a LxCxE motif that recognizes the retinoblastoma protein (Dowdy et al., 1993).

Elucidating how cyclin B1-Cdk1 activity is directed to the right substrate at the right time as cells enter mitosis is essential to understand how cells are remodeled because cyclin B1-Cdk is both the essential trigger and the "workhorse" of mitosis. Evidence for its role as the trigger of mitosis is that mouse embryos with a genetic deletion of cyclin B1 (Brandeis et al., 1998) stop dividing around the four-cell stage as soon as the maternal stock of cyclin B1 is degraded (Strauss et al., 2018); these cells arrest in G2 phase and are unable to initiate mitosis (Strauss et al., 2018). To ensure that cells remain in mitosis, cyclin B1-Cdk1 phosphorylates and activates the Greatwall protein kinase, which generates an inhibitor of the PP2A-B55 phosphatase that antagonizes cyclin B1-Cdk1 in interphase (Castilho et al., 2009; Gharbi-Ayachi et al., 2010; Mochida et al., 2010). In its role as the workhorse of mitosis (Nigg, 1995), cyclin B1-Cdk1 phosphorylates structural components throughout the cell including the

.................................................................................................................................................................................................................................

The Institute of Cancer Research, London, UK.

*M. Jackman and C. Marcozzi contributed equally to this paper; Correspondence to Jonathon Pines: jon.pines@icr.ac.uk; A.L. Tyson's present address is Sainsbury Wellcome Centre, London, UK.

nuclear lamins (Heald and McKeon, 1990; Peter et al., 1990), nuclear pore components (Linder et al., 2017), condensins (Hirano, 2012), and cytoskeletal regulators such as the Rho Guanine nucleotide exchange factor ECT2 (Tatsumoto et al., 1999). Microtubule motors, nonmotor microtubule-associated proteins, endoplasmic reticulum, and Golgi apparatus components are also extensively phosphorylated by cyclin B1-Cdk1 (Champion et al., 2017; Wieser and Pines, 2015). A crucial role for cyclin B1-Cdk1 is to activate the anaphase promoting complex/cyclosome (APC/C), the ubiquitin ligase that will subsequently degrade cyclin B1 itself (Fujimitsu et al., 2016; Golan et al., 2002; Lu et al., 2014; Passmore et al., 2005; Qiao et al., 2016). But cyclin B1-Cdk is also required for the spindle assembly checkpoint (SAC; D'Angiolella et al., 2003; Morin et al., 2012; Vázquez-Novelle et al., 2014; Wieser and Pines, 2015; Hayward et al., 2019) that keeps the APC/C from degrading cyclin B1 (and the separase inhibitor, securin) until all the kinetochores are attached to the mitotic spindle, which is essential for genomic stability (Lara-Gonzalez et al., 2012; Musacchio and Salmon, 2007).

One insight into how one kinase can coordinate so many different events is that cyclin B1-Cdk1 is targeted to different structures as the cell enters mitosis. Human cyclin B1-Cdk is activated on centrosomes (Jackman et al., 2003), and a large fraction immediately moves into the nucleus over ~20 min preceding the breakdown of the nuclear envelope (NEBD; Gavet and Pines, 2010; Hagting et al., 1999; Pines and Hunter, 1991). Subsequently, cyclin B1-Cdk binds to the microtubules around the spindle caps, to chromosomes in early mitosis, and to unattached kinetochores (Bentley et al., 2007; Chen et al., 2008; Pines and Hunter, 1991). These observations indicate that the localization of cyclin B1-Cdk1 may be an important determinant of how specific substrates are recognized at specific times.

The remodeling of the cell at mitosis raises another important question: when interphase macromolecular machines are disassembled in mitosis, do their components, or subcomplexes of their components, contribute to the function of newly assembled mitotic machines? For example, there is an intriguing connection between the NPC and kinetochores because when the NPC is disassembled at the end of prophase, several NPC components relocalize to kinetochores in mitosis, including the Nup107-160 complex (Loïodice et al., 2004; Zuccolo et al., 2007), Nup358/RanBP2, and Crm1 (Dasso, 2006; Joseph et al., 2004; reviewed in Forbes et al., 2015). Moreover, the MAD1 and MAD2 SAC proteins are prominently bound to the NPC in interphase (Chen et al., 1998; Lee et al., 2008); in mitosis, these bind to unattached kinetochores to generate the mitotic checkpoint complex (MCC, composed of MAD2, BubR1, Bub3, and Cdc20) that inhibits the APC/C to prevent premature sister chromatid separation and aneuploidy (London and Biggins, 2014; Moyle et al., 2014; Sudakin et al., 2001; Izawa and Pines, 2015).

In budding yeast, which maintain a nuclear envelope during mitosis, MAD1 modulates nuclear transport in response to kinetochore detachment from microtubules (Cairo et al., 2013), but it is unclear whether MAD1 and MAD2 have an interphase role at the NPC in cells that break down their nuclear envelope in mitosis. Rodriguez-Bravo et al. (2014) have proposed that in interphase, the MAD1/MAD2 heterodimer at the nuclear pore in human cells can catalyze the production of the MCC before kinetochores are assembled from late G2 into mitosis (Gascoigne and Cheeseman, 2013), and that this is required to generate sufficient MCC in interphase and inhibit the APC/C when it is fully activated at NEBD (Rodriguez-Bravo et al., 2014). The importance of this is uncertain, however, since there are several other mechanisms that keep the APC/C in check in interphase: the Emi1 inhibitor and cyclin A-Cdk complexes both inhibit the Cdh1 coactivator (Reimann et al., 2001; Sørensen et al., 2001; Di Fiore and Pines, 2007; Frye et al., 2013), and phosphorylation by cyclin A-Cdk complexes prevents the Cdc20 coactivator from binding the APC/C (Hein and Nilsson, 2016; Labit et al., 2012). Moreover, Cdc20 cannot bind to the APC/C until an autoinhibitory loop of the APC1 subunit is phosphorylated by cyclin-Cdk and Plk1 kinases at mitosis (Qiao et al., 2016; Zhang et al., 2016). A recent report has proposed a further "timer" mechanism that is activated after NEBD, whereby phosphorylation of the Bub1 protein by Cdk1 and the MPS1 kinase recruits MAD1 to kinetochores to generate the MCC independently of the pathway that responds to microtubule attachment (Qian et al., 2017). Thus, the means by which the cell ensures that newly activated mitotic APC/C is kept inhibited until all kinetochores attach to microtubules is a matter of some debate.

Here we uncover a connection between disassembly of the NPC and the generation of a kinetochore competent for SAC signaling that depends on the targeting of cyclin B1 to MAD1. We show that the MAD1 protein binds to cyclin B1 through an acidic patch in a predicted helical domain of MAD1, and that binding is required to recruit cyclin B1 to unattached kinetochores. We further show that cyclin B1 binding to MAD1 is important for the proper release of MAD1 from the nuclear pore and its timely recruitment to kinetochores before NEBD, and thus for the ability of kinetochores to generate a robust SAC signal in early mitosis. Our findings provide evidence for the importance of localized cyclin B1-Cdk1 activity in the coordinated reorganization of the cell as it enters mitosis. We also provide a mechanism for how the cell coordinates activation of the APC/C with the generation of the MCC to keep the APC/C in check in early mitosis and consequently contribute toward genomic stability.

## Results

### Cyclin B1 binds to MAD1 through the acidic face of a helix encoded by exon 4

We sought to understand what controls the highly dynamic behavior of cyclin B1-Cdk1 complexes as the cell enters mitosis; in particular, how cyclin B1 is recruited to specific places in the cell at specific times. To identify binding partners, we immunoprecipitated cyclin B1 from both normal diploid retinal pigment epithelial (RPE) cells and from transformed HeLa cells and analyzed the coprecipitating proteins by mass spectrometry (Table 1 and data submitted to the Proteomics Identification Database (PRIDE) database, PXD017202). We found MAD1 as one of the most prominent proteins in immunoprecipitates from both cell lines. We confirmed MAD1 as a major cyclin B1–binding partner by immunoblotting. It coimmunoprecipitated with

**Table 1.** Relevant proteins significantly enriched after CCNB1 immunoprecipitation and quantitative mass spectrometry analysis

| Accession no.: | Description | IgG | Mean abundance | | | FC (Adjusted P value) | | |
|---|---|---|---|---|---|---|---|---|
| | | | CCNB1 | CCNB1 | CCNB1 | CCNB1 | CCNB1 | CCNB1 |
| WT | D2 | B12 | WT/IgG | WT/D2 | WT/B12 | | | |
| P14635 | Cyclin B1 | 737.6 | 60,035.3 | 45,962.5 | 54,126.8 | 6.60 (1.61 e-07) | −0.54 (0.96) | −0.03 (0.99) |
| Q9Y6D9 | MAD1 | 1,170.7 | 50,184.2 | 1,280.1 | 3,631.2 | 5.47 (2.39 e-06) | −5.25 (3.06 e-05) | −3.54 (0.016) |
| Q15013 | MAD2 | 66.1 | 2,914.0 | 60.9 | 233.5 | 5.68 (1.02 e-05) | −5.61 (5.66e-05) | −3.83 (0.019) |

Mean abundance is the average abundance from three biologically independent experiments, and FC is the fold change. Significantly enriched proteins were identified by LIMMA test (P < 0.05).

cyclin B1 from cells in both G2 phase and mitosis (Fig. 1 A; see also Fig. S1), but we noticed that there were two isoforms of MAD1 detected by immunoblotting HeLa cell lysates, of which only the more slowly migrating form (MAD1α) coimmunoprecipitated with cyclin B1 (Fig. 1 A). A previous study had identified an alternatively spliced form of MAD1 (MAD1β) in hepatocellular carcinoma cells (Sze et al., 2008), which lacks the 47 amino acid–encoding exon 4 of MAD1α and migrates at the same molecular mass as our faster migrating form of MAD1. Therefore, we expressed MAD1α and MAD1β from cDNAs and found that only MAD1α bound to cyclin B1 (Fig. S1 C). In agreement with this, expressing a series of truncation mutants showed that residues 39–329 were able to bind to cyclin B1 (data not shown).

These analyses implicated the peptide sequence encoded by exon 4 (residues 51–97) as important for binding to cyclin B1; however, exon 4 also contains the nuclear localization sequence (79KKR82) of MAD1, previously shown to be important for its function and its proper localization to the NPC in interphase (Sze et al., 2008). Therefore, we sought to narrow down the residues required to interact with cyclin B1. A region of the RepoMan protein between amino acids 403–550 had been reported to bind cyclin B1 (Qian et al., 2015); when we compared this region to exon 4 of MAD1, we found a five amino acid region of homology (L51-E56aa of MAD1, Fig. 1 B). This region of MAD1 is predicted to be part of helical region by JPred (Cole et al., 2008), and likely to form a coiled-coil. Mutating all five residues (L51G/E52A/E53A/R54A/E56A) disrupted the ability of MAD1 to bind to the nuclear envelope (Fig. S1, D and E), likely due to altering the ability of MAD1 to form a coiled-coil structure at this position; therefore, we sought a more refined mutation that still maintained the hydrophobic (L51, A55) and charged (E52, R54) residues predicted to stabilize the coiled-coil structure. We identified two acidic residues within this region, E53 and E56, that should be exposed to the outside face of the predicted coiled-coil (Fig. S1 D). An interaction with an acidic surface could conceivably be used to confer specificity for binding to B-type cyclins in animal cells because comparing the structures of B- and A-type cyclins shows that B-type cyclins are distinguished by their conserved basic patches at their interface with Cdk2 (Brown et al., 2007). When we made the double charge substitution of E53/56K, we found this severely perturbed binding of MAD1 to cyclin B1 in vitro (Fig. 1, C and D) but

not localization of full length MAD1 to the nuclear envelope in interphase (Fig. S1 E), nor binding to MAD2 (see below).

## MAD1 recruits cyclin B1 to kinetochores

We sought to identify the function of the binding between MAD1 and cyclin B1. We used CRISPR/Cas9D10A (Fig. S2 A) to introduce the E53/56K mutation into both alleles of MAD1 in RPE1 cells in which we had tagged one allele of cyclin B1 with the Venus yellow fluorescent protein, and one allele of MAD2 with the Ruby red fluorescent protein. We isolated single cell clones of the parental and mutant cells and confirmed the MAD1 point mutations in two independent clones (7D2 and 8B12) by PCR analysis and genome sequencing (Fig. S2 B). In agreement with our in vitro analysis, the MAD1 E53/56K mutants were unable to bind cyclin B1 (Fig. 2 A and Table 1), but were still able to bind to MAD2 (Fig. 2 B), as expected, because MAD2 binds to a region of MAD1 450 amino acids away. Mutating MAD1 in the RPE1 cyclin B1-Venus:Ruby-MAD2 cells allowed us to assay cyclin B1 and MAD2 recruitment to kinetochores in living cells. This showed that in both clones, the E53/56K mutation prevented cyclin B1 but not MAD2 from being recruited to unattached kinetochores in prometaphase cells (Fig. 2 C and Video 1, Video 2, and Video 3). We were thus in a position to determine the role of MAD1 binding to cyclin B1 and recruiting it to kinetochores.

## MAD1 binding to cyclin B1 is required for genomic stability

We first analyzed the chromosomal stability of our MAD1 E53/56K clones compared with the parental RPE1 cells by counting the chromosome number in metaphase spreads. This showed that mutating MAD1 dramatically increased chromosomal instability: more than 85% of cells in both mutant clones had gained or predominantly lost chromosomes, compared with 24% of the parental (Fig. 3 A). The increase in chromosomal instability in the MAD1 E53/56K clones might be explained by a weaker SAC. To test this, we assayed the SAC under three conditions: untreated cells, where the time from NEBD to anaphase is determined by the SAC (Fig. 3 B); mitotic delay in cells treated with low doses of nocodazole (Fig. 3 C); and mitotic delay in cells treated with paclitaxel (Fig. S3). There was no significant difference in timing between the wild-type and mutant clones in untreated cells, but neither MAD1 mutant clone was able to arrest for as long as the parental cells in response to nocodazole or paclitaxel. Thus, we conclude that MAD1 recruitment of cyclin

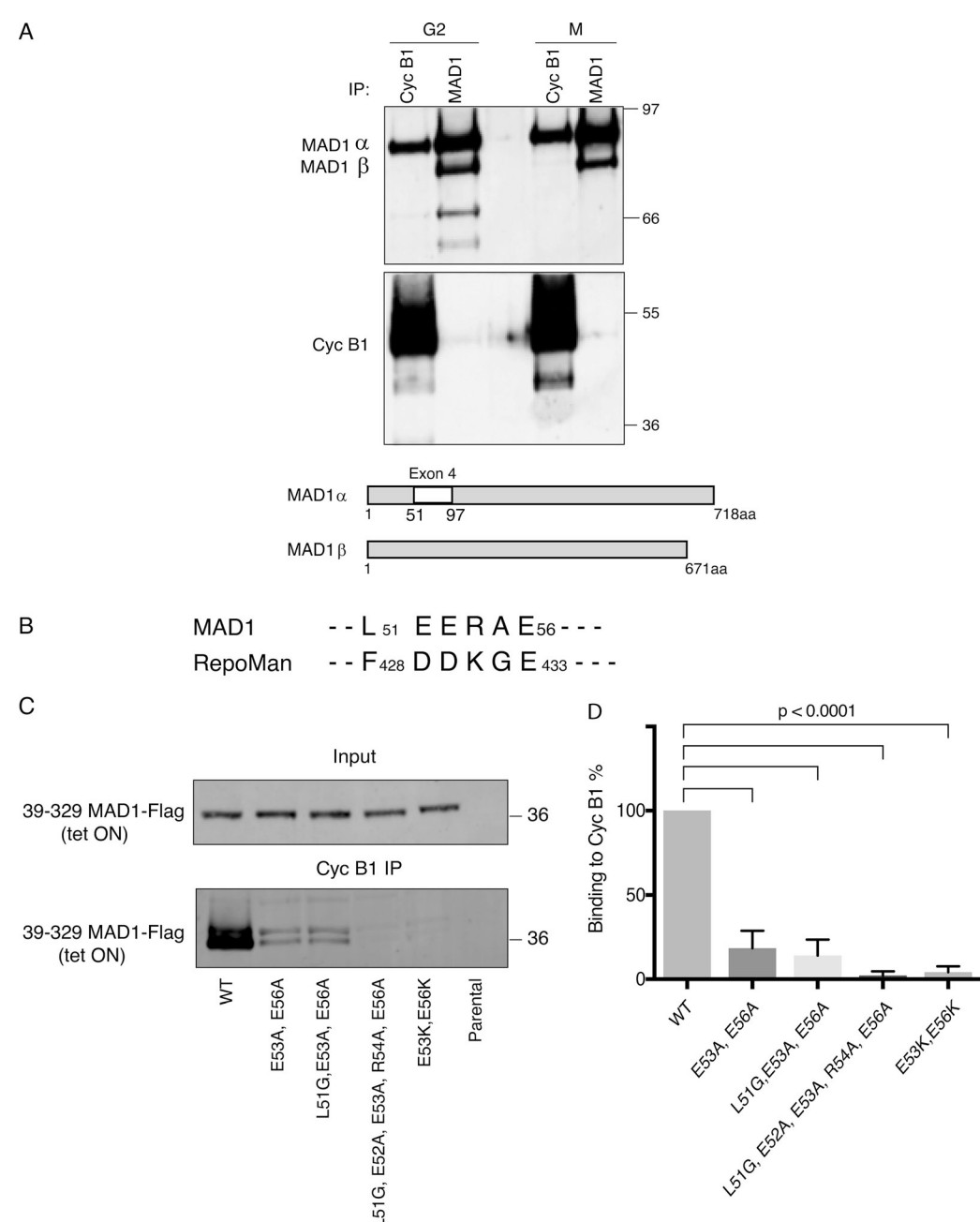

Figure 1. **MAD1 binds to cyclin B1 through the acidic face of a helix within exon 4. (A)** HeLa cells were synchronized in either G2 phase or mitosis (M), cyclin B1 and MAD1 were immunoprecipitated (IP), subjected to SDS-PAGE and immunoblotted with anti-MAD1 (upper panel), and anti-cyclin B1 antibodies (lower panel). The difference in stoichiometries of binding is likely to be related to epitope accessibility, but we note that Schweizer et al. (2013) previously identified cyclin B1 in MAD1 immunoprecipitates. Schematic shows the location of exon 4 in MAD1α that is absent from MAD1β. **(B)** Similarity between MAD1 and Repoman sequences within the regions found to interact with cyclin B1. **(C)** HeLa cells expressing wild-type or mutated MAD1-Flag (39-329aa) from a tetracycline-inducible promoter (tet ON) were synchronized in either G2 phase or mitosis 12 h after adding tetracycline. Cyclin B1 was immunoprecipitated, subjected to SDS-PAGE, blotted with anti-FLAG antibody, and assayed on a LiCOR Odyssey scanner. **(D)** The data from three experimental repeats were normalized to the amount of cyclin B1 binding to wild-type MAD1 and plotted using Prism software. Error bars represent SD; two-tailed P values were calculated using an unpaired *t* test.

B1 to kinetochores strengthens the SAC and that this contributes to long-term genomic stability.

### Cells with MAD1 mutants that cannot bind cyclin B1 are sensitive to partial inhibition of MPS1

In our previous studies on the strength of SAC signaling, we showed that there was an inverse correlation between the strength of the SAC and the dose of an MPS1 inhibitor (Collin et al., 2013). We reasoned that partial inhibition of MPS1 might sensitize cells and uncover a more penetrant role for cyclin B1-Cdk1 binding to MAD1. In agreement with this, when we treated cells with low doses of an MPS1 inhibitor, either reversine (Fig. 4) or AZ3146 (Fig. S4), we found that the SAC was much more severely compromised in the MAD1 E53/56K mutant

A

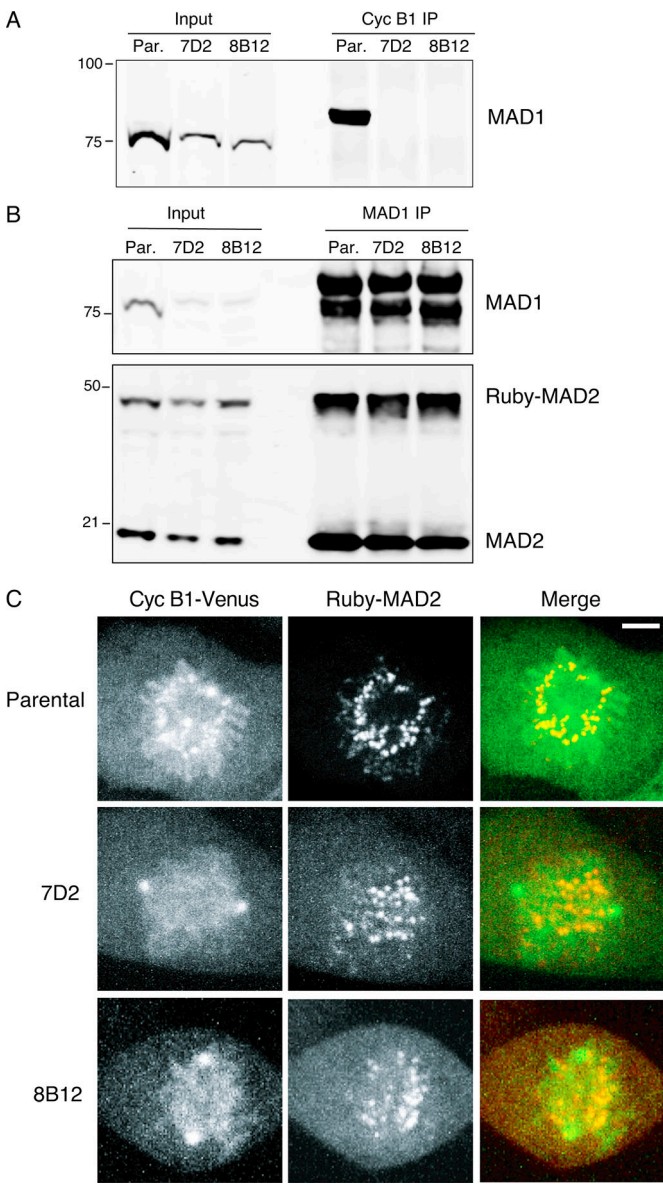

Figure 2. **The E53K/E56K mutation prevents MAD1 binding to cyclin B1 but not MAD2. (A)** Parental RPE cyclin B1-Venus$^{+/-}$:Ruby-MAD2$^{+/-}$ cells (Par.) or clones 7D2 and 8B12 carrying a homozygous mutation of E53/E56K in MAD1 were synchronized to enrich for G2 phase and mitosis; cyclin B1 (Cyc B1) was immunoprecipitated, subjected to SDS-PAGE, and immunoblotted with anti-MAD1 antibodies. **(B)** Parental RPE cyclin B1-Venus$^{+/-}$:Ruby-MAD2$^{+/-}$ cells (Par.) or clones 7D2 and 8B12 were synchronized in G2 and M phase, and MAD1 was immunoprecipitated, and immunoblotted with anti-MAD1 (upper panel) and anti-MAD2 (lower panel) antibodies. **(C)** Parental RPE cyclin B1-Venus$^{+/-}$:Ruby-MAD2$^{+/-}$ cells or MAD1 E53/E56K clones 7D2 and 8B12 were assayed by spinning disk confocal time-lapse microscopy. Images of maximum intensity projections of cyclin B1-Venus (left, green), Ruby-MAD2 (middle, red), and the merged image (right) for a representative prometaphase cell are shown. See Videos 1, 2, and 3. Data shown for all panels are representative of three independent experiments. Total number of cells: parental n = 21 cells, clone 7D2 n = 16 cells, clone 8B12 n = 10 cells. Scale bar, 3 µm.

clones than in the parental clones, as assayed by both the timing from NEBD to anaphase in the absence of a spindle poison (Fig. 4 A) and by the ability of cells to arrest in nocodazole (Fig. 4 B).

Furthermore, live-cell analyses of chromosome behavior in these cells revealed that around 80% of the MAD1 mutant cells failed to form a proper metaphase plate and performed anaphase with a large number of lagging chromosomes, compared with <20% of the parental cells (Fig. 4 C; and see Videos 4, 5, 6, and 7).

The premature sister chromatid separation exhibited by the MAD1 mutant cells could have one of two explanations: either they were unable to activate the SAC, or they were unable to maintain the SAC while the number of signaling kinetochores diminished following microtubule attachment. To distinguish between these two possibilities, we analyzed the kinetics of cyclin B1-Venus degradation (Fig. 4 D). In parental cells, cyclin B1-Venus was stable until the SAC was inactivated in metaphase, but in both MAD1 mutants, it was only stable for a few minutes after NEBD and was then degraded at a similar rate to wild-type cells that have satisfied the SAC. This showed that the SAC was initially active in the mutant cells but could not be maintained (Fig. 4 D). Thus, we conclude that the SAC is much more dependent on MPS1 kinase activity when MAD1 is unable to recruit cyclin B1.

## MAD2 recruitment to kinetochores is delayed when MAD1 cannot bind cyclin B1

To gain insight into the mechanism underlying the weaker SAC signaling and the greater dependence on MPS1 in cells where MAD1 cannot bind cyclin B1, we studied the recruitment of SAC proteins to kinetochores as cells began mitosis. To do this, we used CRISPR/Cas9$^{D10A}$ to introduce an RFP670 fluorescent tag into the MIS12 protein (Fig. S5, A–C; see also Videos 6 and 7) so that we could identify kinetochores in living cells. We then assayed the recruitment of MAD2 to kinetochores by quantifying the colocalization between MAD2 and MIS12 (see Materials and methods). This showed a striking difference between the parental and MAD1 E53/56K mutant clones (Fig. 5, A and B). In parental cells, MAD2 began to be recruited to the newly formed kinetochores 10 min or more before NEBD, whereas recruitment was markedly delayed in the MAD1 mutant cells; indeed, in the cells of one clone (7D2), MAD2 was not recruited until NEBD. In cells of the other clone (8B12), MAD2 recruitment was also much slower than normal, and never reached the amounts seen in parental cells (Fig. 5 B). Thus, we conclude that kinetochores in MAD1 E53/56K mutant cells are compromised in their SAC signaling due to delayed recruitment of MAD2.

## MAD1 remains associated with TPR and condensing chromosomes when it cannot bind cyclin B1

MAD2 has to bind MAD1 to be recruited to kinetochores (Chen et al., 1998); therefore, we analyzed the behavior of MAD1 in mutant and parental cells as they entered mitosis. We found that like MAD2, MAD1 recruitment to kinetochores was also perturbed when MAD1 was unable to bind cyclin B1, and this was because the MAD1 now appeared to remain associated with the condensing chromosomes (Fig. 6 A). We hypothesized that this might be caused by inefficient release from the nuclear basket; therefore, we analyzed the behavior of the TPR protein that binds MAD1 and is required for its localization to the NPC (Lee et al., 2008). In agreement with our hypothesis, this revealed

Figure 3. **MAD1 binding to Cyclin B1 is required for genomic stability. (A)** Chromosome number per cell was assayed for parental RPE yclin B1-Venus$^{+/-}$: Ruby-MAD2$^{+/-}$ cells and the MAD1 E53/E56K clones 7D2 and 8B12 by metaphase spreads in three independent experiments. Black dots indicate 46 chromosomes and n indicates the number of cells assayed. Parental n = 89 cells, clone 7D2 n = 74 cells, clone 8B12 n = 80 cells. **(B)** The time from NEBD to anaphase was measured for parental RPE cyclin B1-Venus$^{+/-}$:Ruby-MAD2$^{+/-}$ cells and the MAD1 E53/E56K clones 7D2 and 8B12 by time-lapse DIC microscopy. Violin plots show the data from three experiments (median time indicated by the black bar); n indicates the total number of cells analyzed. Parental n = 198 cells, clone 7D2 n = 197 cells, clone 8B12 n = 201 cells. **(C)** The duration of the mitotic arrest for parental RPE cyclin B1-Venus$^{+/-}$:Ruby-MAD2$^{+/-}$ cells and the MAD1 E53/E56K clones 7D2 and 8B12 was assayed by time-lapse DIC microscopy in 55 nM nocodazole. Data are plotted as for B; n indicates the total number of cells analyzed. Parental n = 273 cells, clone 7D2 n = 228 cells, clone 8B12 n = 231 cells. The two-tailed P values were calculated using a Mann–Whitney unpaired t test.

that mutant MAD1 that cannot bind cyclin B1 remained associated with TPR on chromosomes in early mitosis (Fig. 6 A).

We then asked whether the release of MAD1 from TPR might also be sensitive to MPS1 kinase activity and analyzed the localization of TPR and MAD1 in cells treated with a low dose of an MPS1 inhibitor. This had a striking effect in the MAD1 E53/56K cells where TPR and MAD1 almost completely colocalized around the chromosomes and very little MAD1 was able to bind to the newly formed kinetochores. We saw a similar but milder effect on MAD1 in the parental RPE cells (Fig. 6 B).

**The effect of the MAD1 mutant is partially rescued by directing cyclin B1 to the NPC**

Our results indicated that localized cyclin B1-Cdk1 synergizes with MPS1 to release MAD1 from TPR and allow its recruitment to kinetochores. If this was the case, then we should be able to detect cyclin B1 colocalizing with TPR in wild-type cells, and this should be perturbed in the MAD1 mutants. To assay this, we fixed cells, stained them with antibodies to detect cyclin B1 and TPR, and identified prophase cells by the extent of chromosome condensation and the nuclear import of cyclin B1. This analysis

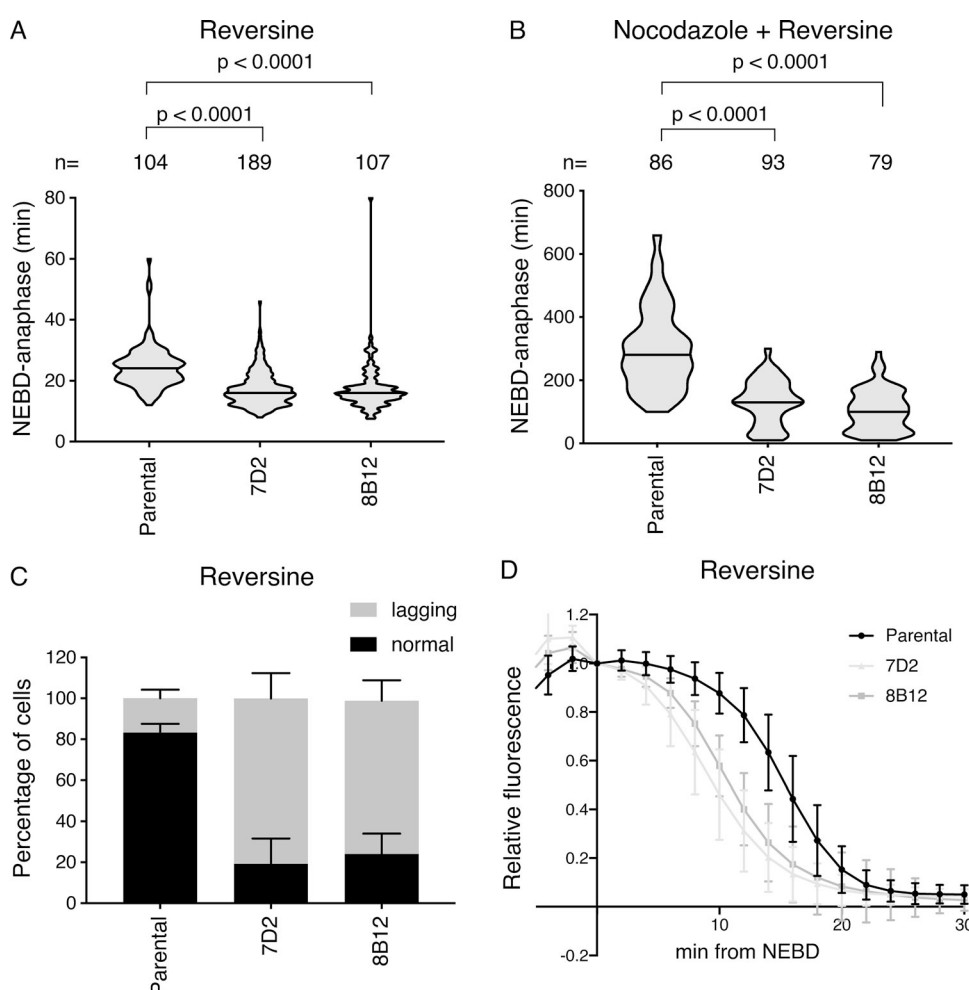

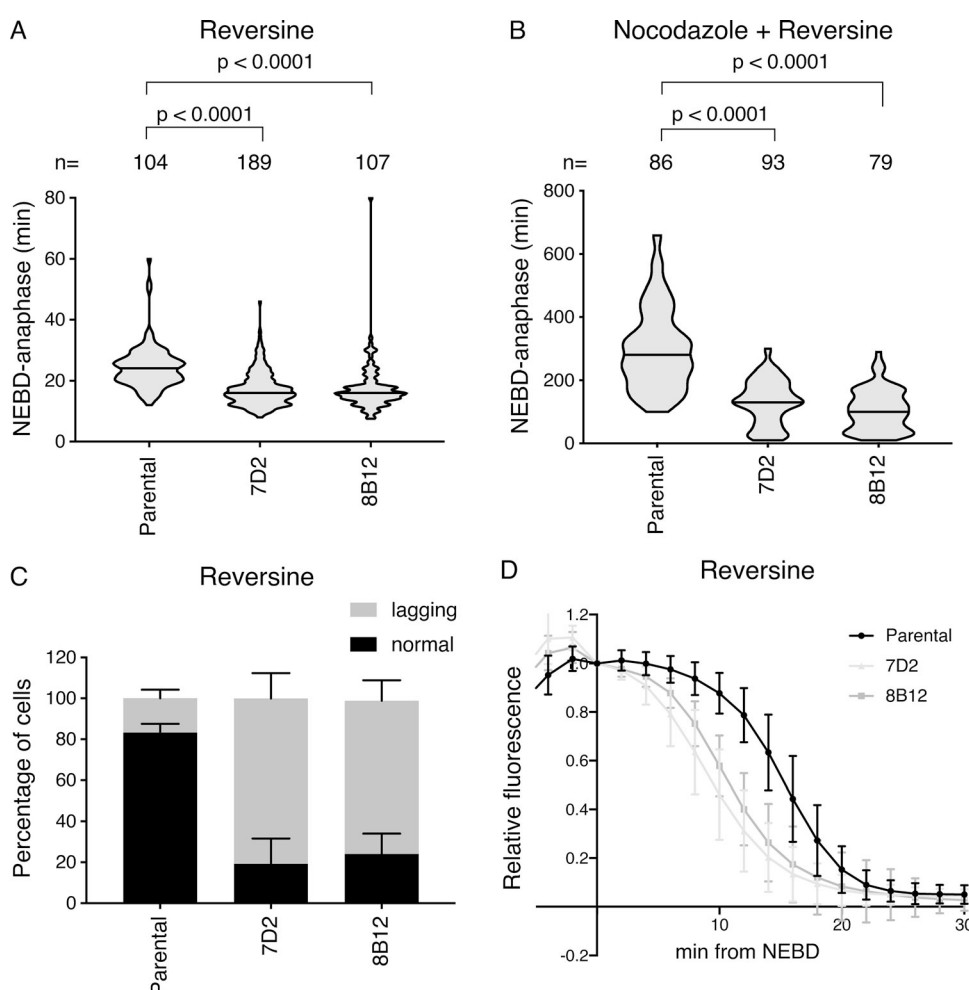

Figure 4. **Cells with mutant MAD1 mutant that cannot bind cyclin B1 are sensitive to partial MPS1 inhibition**. **(A and B)** The timing from NEBD to anaphase was measured in parental RPE cyclin B1-Venus$^{+/-}$:Ruby-MAD2$^{+/-}$ cells and in the MAD1 E53/E56K clones 7D2 and 8B12 by time-lapse DIC microscopy. Cells were treated with 166 nM reversine (A) or with 55 nM nocodazole plus 166 nM reversine (B) and the data plotted as in Fig. 3. Median values are shown as a black line, n indicates the number of cells analyzed, and the two-tailed P values were calculated using an unpaired Mann–Whitney t test. Data from at least three independent experiments are shown. **(A)** Parental n = 104 cells, clone 7D2 n = 189 cells, clone 8B12 n = 107 cells. **(B)** Parental n = 86 cells, clone 7D2 n = 93 cells, clone 8B12 n = 79 cells. **(C)** Quantification of lagging chromosomes in 166 nM reversine-treated parental RPE cyclin B1-Venus$^{+/-}$, Ruby-MAD2$^{+/-}$ MAD1 wild type (wt) and the MAD1 E53/E56K 7D2 and 8B12 clones stained with SiR-DNA (see Videos 4 and 5). Error bars show SD. Parental n = 110 cells, clone 7D2 n = 182, clone 8D2 n = 108. **(D)** Cyclin B1-Venus degradation curves from parental RPE cyclin B1-Venus$^{+/-}$:Ruby-MAD2$^{+/-}$ clones and from the MAD1 E53/E56K clones 7D2 and 8B12 treated with 166 nM reversine. Cyclin B1-Venus fluorescence levels were normalized to the value at NEBD and the mean values from >20 cells analyzed per experiment plotted, with the bars showing the SD. Data are representative of three experiments.

showed that a proportion of cyclin B1 colocalized with TPR at the nuclear envelope in wild-type cells (Fig. 7 A), and that this was significantly reduced in cells with mutated MAD1 (Fig. 7 A, quantified in Fig. 7 B). We postulated that restoring cyclin B1 to the nuclear pore might rescue the effect of the MAD1 E53/56K mutant on the SAC. To test this, we took the POM121 nuclear pore protein that binds the inner nuclear membrane, and fused an mTurquoise2 (mTurq2)-labeled GFP-binding protein nanobody (GBP) to its C terminus. This should bind to cyclin B1-Venus and recruit it to the NPC. We randomly integrated the cDNA encoding this fusion protein into the RPE1 cyclin B1-Venus:Ruby-MAD2 cell line and into MAD1 E53/56K clone 7D2 where MAD2 was only released from the NPC at NEBD. Live-cell imaging revealed that cells expressing the POM121-mTurq2-GBP fusion protein recruited cyclin B1-Venus to the NPC (Fig. 7 C).

We then treated these cells with 100 nM paclitaxel plus 166 nM reversine and assayed the ability of these cells to maintain a mitotic arrest. We compared the behavior of these cells to cells expressing randomly integrated POM121 fused to mTurq2 alone and to the parental cyclin B1-Venus:Ruby-MAD2 cells expressing either POM121 or POM121-GBP fusion proteins as controls (Fig. 7 D). In four separate experiments, we found that cells expressing the POM121-GBP fusion protein were able to maintain a mitotic arrest for more than twice as long as cells expressing POM121, and live-cell imaging showed that in 10 out of 10 of these cells, MAD2 was released from the nuclear envelope 2 to 4 min earlier than NEBD (see Videos 8 and 9). Thus, we conclude that cyclin B1-Cdk1 at the NPC is required for the timely release of MAD1 from the NPC, which contributes toward the generation of a robust SAC.

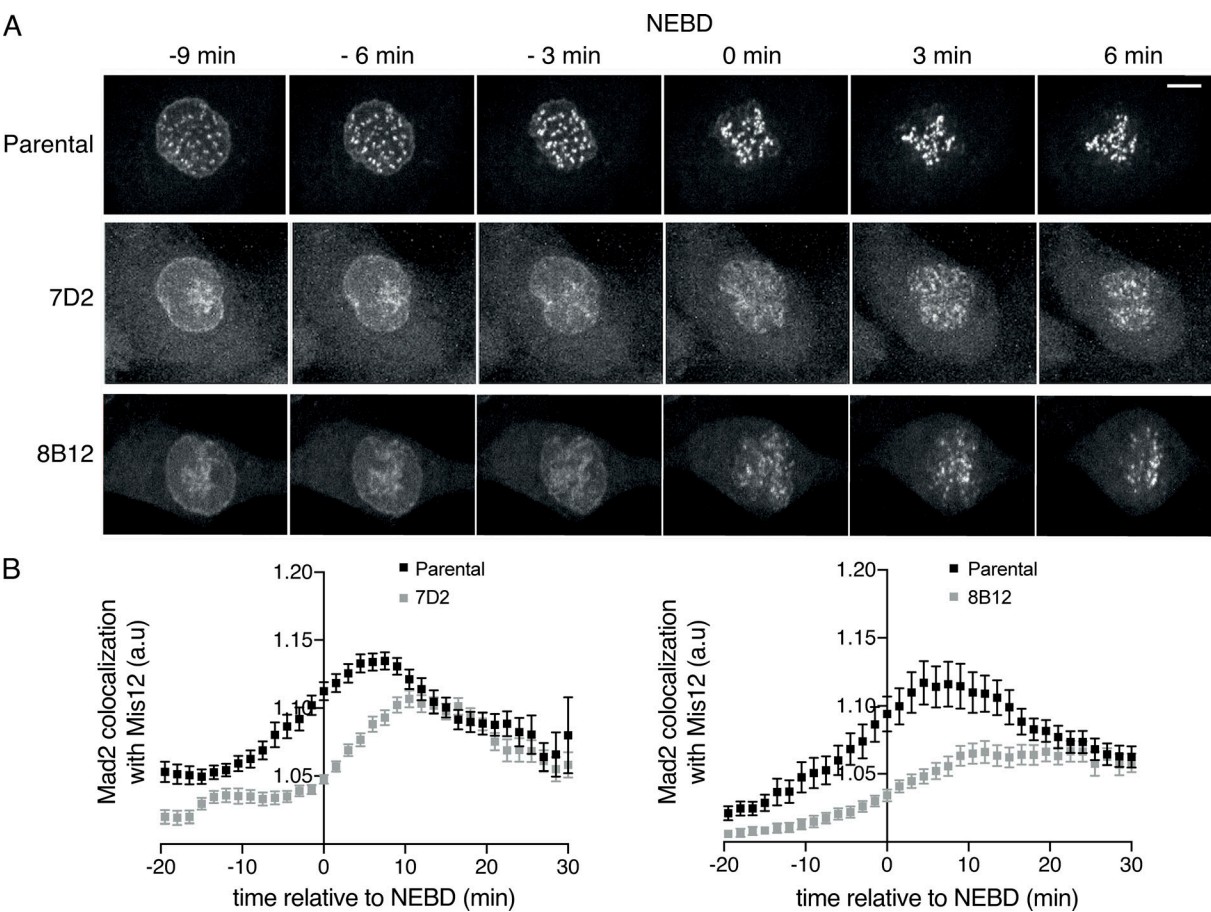

Figure 5. **MAD2 recruitment to kinetochores is delayed when MAD1 cannot bind cyclin B1. (A)** Maximum intensity projections at the indicated times from time-lapse fluorescence of parental RPE cyclin B1-Venus$^{+/-}$:Ruby-MAD2$^{+/-}$:RFP670-MIS12$^{+/+}$ cells and the MAD1 E53/E56K$^{+/+}$: RFP670-MIS12$^{+/+}$ clones 7D2 and 8B12 showing Ruby-MAD2 localization relative to NEBD. Scale bar, 5 μm. **(B)** Quantification of Ruby-MAD2 colocalization with RFP670-MIS12 using the Coloc-3DT program (see Materials and methods) relative to NEBD. Graphs show values obtained from at least 40 cells for each clone in three independent experiments. Error bars indicate SEM.

## Discussion

In this study, we have shown that cyclin B1 binds to the MAD1 protein through a predicted acidic patch on a helix of MAD1. Unexpectedly, we find that MAD1 recruits cyclin B1-Cdk1 to promote its own release from the NPC before NEBD, and that cyclin B1-Cdk1 coordinates with the MPS1 kinase to achieve this. The importance of releasing MAD1 before NEBD is that this allows it to bind to kinetochores, where it can begin to generate the MCC to inhibit the APC/C as it is activated by cyclin B1-Cdk1. Thus, our findings identify a simple but elegant mechanism by which the rising level of cyclin B1-Cdk1 activity before NEBD activates the APC/C and sets up the conditions to inhibit it until all the chromosomes attach to the mitotic spindle, thereby contributing to genomic stability.

We identified MAD1 as the most prominent interaction partner of cyclin B1-Cdk1. We and others (Alfonso-Pérez et al., 2019) identified the N terminus of MAD1 as the binding site for cyclin B1. It is intriguing to note that this binding site is lost in the MAD1β spliced form that is prominent in hepatocellular carcinoma cell lines (Sze et al., 2008); it is conceivable that the inability to bind cyclin B1, along with the loss of the nuclear localization signal, might contribute to their genomic instability.

We subsequently narrowed down the cyclin B1 binding motif to a predicted acidic patch on a helical region of MAD1, a region identified independently by Allan et al. (2020). Although beyond the scope of our present study, we are currently determining whether this is a conserved interaction motif for other mitotic substrates of cyclin B1. If so, this will be, to our knowledge, the first interaction motif specific for the major mitotic kinase in animal cells. It is interesting to note that recognition of a helix may be a conserved feature of the cyclins since the D-type cyclins recognize a predicted helix in the C terminus of retinoblastoma protein (Topacio et al., 2019).

It is notable that preventing MAD1 from binding to cyclin B1 perturbs its release from the NPC even though there is plenty of active cyclin B1-Cdk1 freely diffusing within the cell. It is formally possible that cyclin B1 may be acting in a noncatalytic role to promote MAD1 release, for example as a scaffold to recruit another protein, but since both MAD1 and TPR are phosphorylated in mitosis on CDK consensus sites (www.phosphosite.org), and cyclin B1-Cdk1 phosphorylates Nup98 and Nup53 to promote NPC disassembly (Linder et al., 2017), applying Occam's razor would argue that cyclin B1-Cdk1 activity at the NPC helps release MAD1 from the nuclear pore. Thus, our study likely

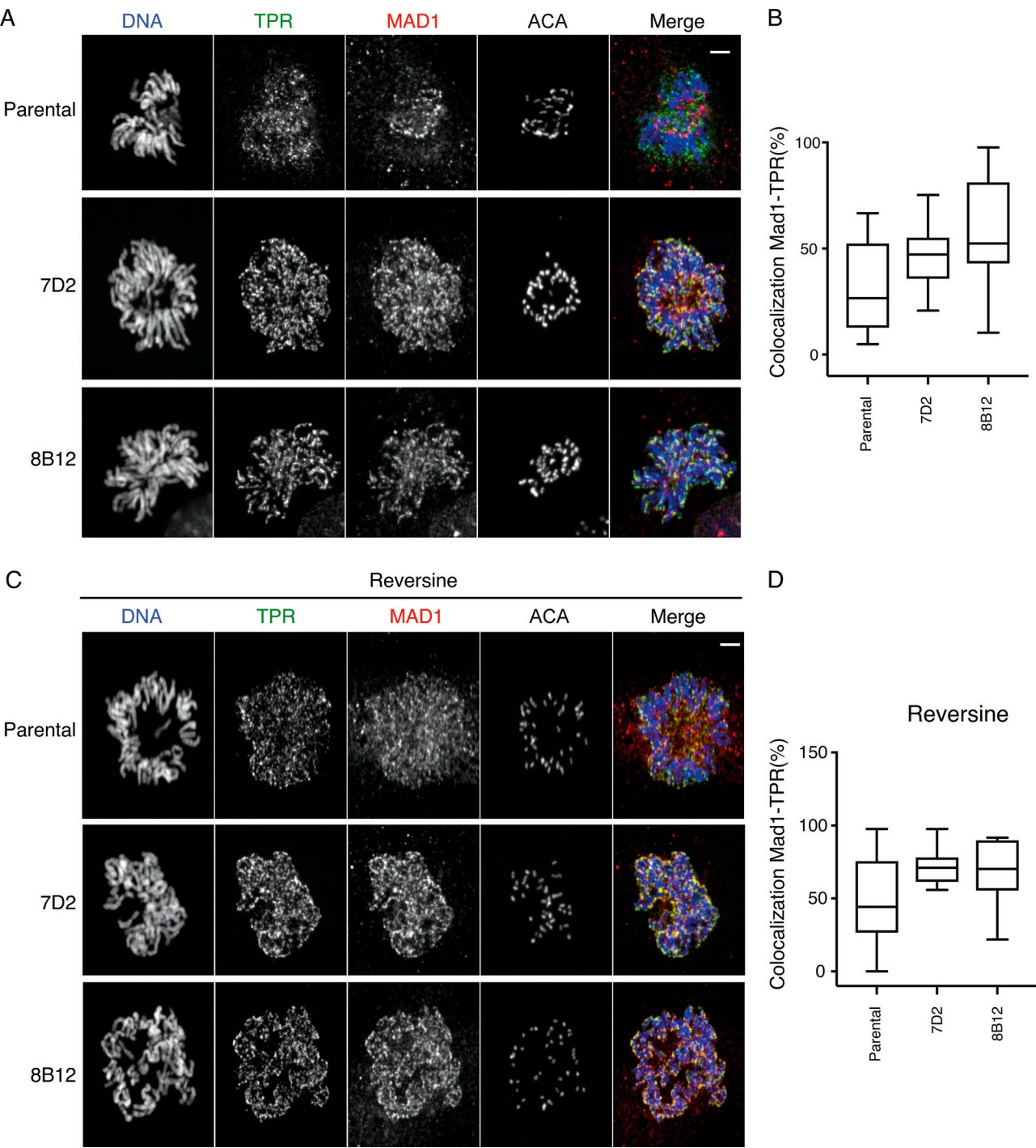

Figure 6. **MAD1 remains associated with TPR and condensing chromosomes when it cannot bind cyclin B1. (A)** Prometaphase parental RPE cyclin B1-Venus$^{+/-}$:Ruby-MAD2$^{+/-}$ cells and the MAD1 E53/E56K clones 7D2 and 8B12 were fixed and stained with Hoechst 33342 (blue), anti-TPR (green), anti-MAD1 (red), and ACA antibodies as indicated. Scale bar, 2 μm. **(B)** Quantification of MAD1-TPR colocalization percentage shown in A (parental $n$ = 12 cells; 7D2 $n$ = 10 cells; 8B12 $n$ = 10 cells). **(C)** Cells were fixed and stained as in A except that they were pretreated with 166 nM reversine. Scale bar, 2 μm. Images are representative of two independent experiments. **(D)** Quantification of MAD1-TPR colocalization percentage shown in C (parental $n$ = 11 cells; 7D2 $n$ = 11 cells; 8B12 $n$ = 9 cells).

identifies an important function for localized cyclin B1-Cdk1 activity and adds to our understanding of the increasing importance of local kinase-phosphatase gradients in controlling the cell (reviewed in Pines and Hagan, 2011). Spatial control of cyclin B1-Cdk1 has been clearly demonstrated in triggering mitosis

from the spindle pole body in fission yeast (Grallert et al., 2013; Hagan and Grallert, 2013), as has the spatial control of Plk1 through its recruitment to substrates previously phosphorylated by Cdk1 (Elia et al., 2003a,b), and in the control of error correction at kinetochores through the balance of Aurora B and PP1/

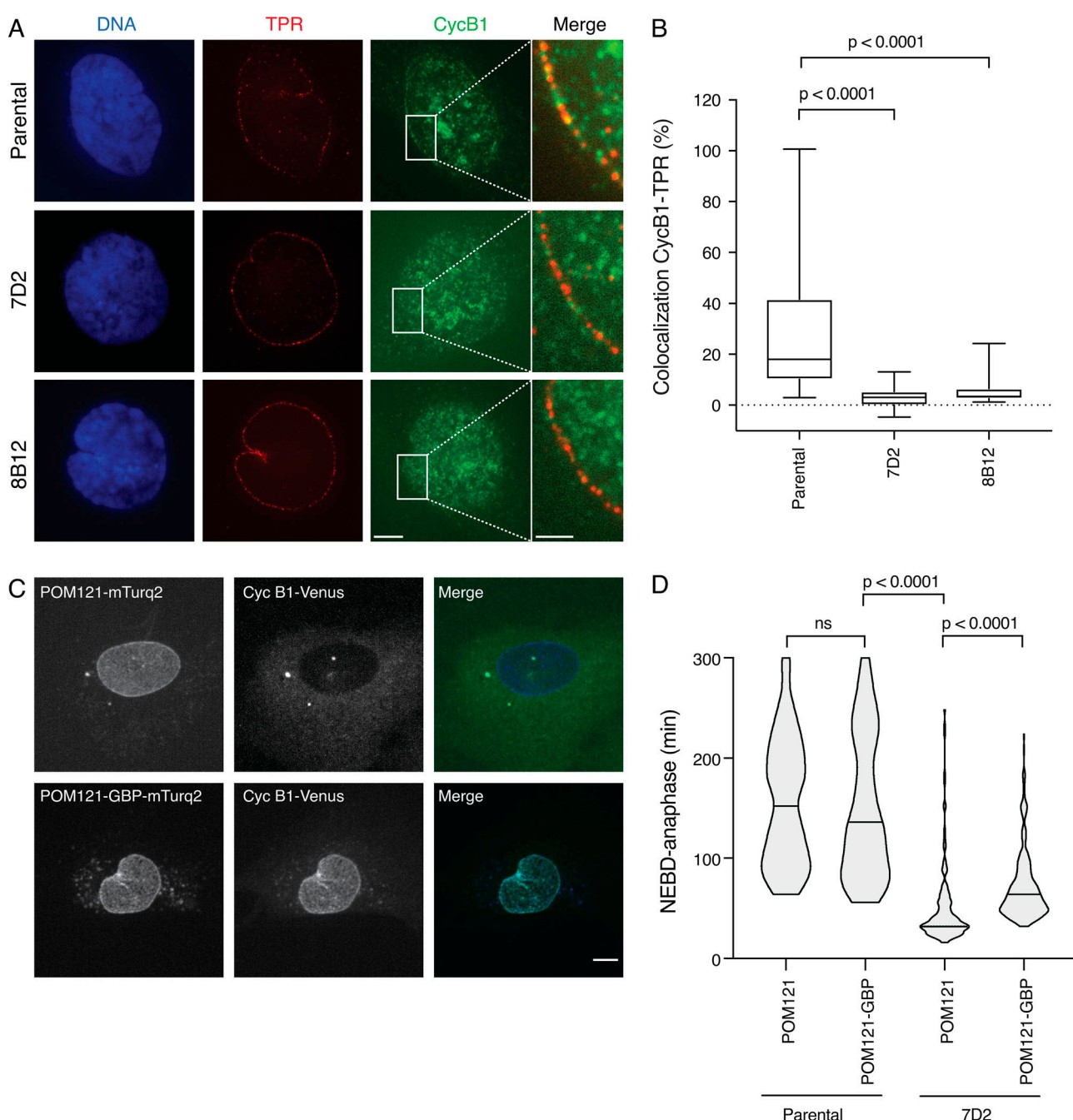

Figure 7. **Cyclin B1 colocalizes with TPR at the nuclear membrane during mitotic entry, and targeting it to the NPC partially restores the SAC.** **(A)** Representative immunofluorescence images of parental RPE cyclin B1-Venus$^{+/-}$:Ruby-MAD2$^{+/-;}$ RFP670-MIS12$^{+/+}$ and MAD1 E53/E56K 7D2 and 8B12 clones fixed and stained for TPR, cyclin B1, and DNA just before NEBD. Scale bar, 5 μm for labeled panels. Right: Merged images of TPR and cyclin B1 in the area highlighted by the white rectangle. Scale bar, 2 μm. **(B)** Quantification of the colocalization between TPR and cyclin B1. Box plots show the mean and quartile values; whiskers show the maximum and minimum values. The results are calculated from at least two independent experiments (parental RPE clone, $n = 34$ cells; 7D2 clone, $n = 24$ cells; 8B12 clone, $n = 24$ cells). **(C)** Maximum projection images of parental RPE cyclin B1-Venus$^{+/-}$:Ruby-MAD2$^{+/-}$:RFP670-MIS12$^{+/+}$ cells expressing either POM121-mTurq2 (top panels) or POM121-GBP-mTurq2 (bottom panels). Left: Localization of POM121-mTurq2 or POM121-GBP-mTurq2 at the NPC; middle: cyclin B1-Venus; right: the merged images. Scale bar, 5 μm. **(D)** The duration of mitotic arrest in 166 nM reversine + 100 nM paclitaxel for parental RPE cyclin B1-Venus$^{+/-}$:Ruby-MAD2$^{+/-}$ cells and the MAD1 E53/E56K clone 7D2 expressing POM121-mTurq2 or POM121-GBP-mTurq2 was assayed by time-lapse microscopy and the data plotted using Prism software. The two-tailed P values were calculated using a Mann–Whitney unpaired $t$ test. For the POM121-GBP-mTurq2-expressing cells, only those cells where cyclin B1-Venus was clearly recruited to the nuclear envelope were analyzed. See Videos 8 and 9.

PP2A phosphatases activities (Liu et al., 2010; Welburn et al., 2010; reviewed in Gelens et al., 2018; Liu et al., 2010).

In addition to emphasizing the importance of spatial control of the mitotic kinases, our study also identifies how the reorganization of the interphase cell is important for the subsequent function of mitosis-specific structures; in particular, how the disassembly of the NPC is coordinated with assembly of a functional kinetochore. The connection between NPC components and the kinetochore has been known for some time: in addition to MAD1/MAD2, the Nup107-160 complex, Nup358/RanBP2, and Crm1 proteins all associate with the kinetochores (Arnaoutov et al., 2005; Joseph et al., 2004; Zuccolo et al., 2007). Our findings now show how the timing of NPC disassembly is important for timely recruitment of MAD1 and MAD2 to kinetochores.

Our work implicating the interaction between TPR and MAD1 as particularly important in the SAC is in line with some previous studies. Human TPR has been reported to bind to kinetochores, where it has been suggested to act as a loading factor to recruit MAD1/MAD2 (Lee et al., 2008); however, in our studies, we have not been able to detect TPR at kinetochores. Indeed, our data indicate that TPR antagonizes MAD1 binding to kinetochores. In agreement with this, *Drosophila*, TPR/Megator has been reported to strengthen the SAC but does not bind kinetochores (Lince-Faria et al., 2009). There is more agreement on a role for the interaction between TPR and MAD1/MAD2 in interphase cells, where in both *Drosophila* and human cells, TPR is important to stabilize MAD1 and MAD2 protein levels (Schweizer et al., 2013; Lee et al., 2008), perhaps through preventing MAD1 SUMOylation (Schweizer et al., 2013).

The role of Cdk1 in NPC disassembly has been most clearly shown in studies using a powerful in vitro system (Linder et al., 2017; Marino et al., 2014). These studies have implicated the Plk1 and Nek kinases working in coordination with Cdk1 to phosphorylate the core NPC components Nup98 and Nup53. Moreover, a recently published study revealed an important role for the MPS1 kinase in releasing MAD1 and MAD2 from the nuclear pores through phosphorylating TPR in *Drosophila* (Cunha-Silva et al., 2020). Our study indicates that this is likely to be conserved in evolution, and reveals that MPS1 is coordinated with Cdk1 in freeing MAD1 from TPR at the inner-NPC "basket."

MPS1 has also been shown to localize to the nuclear pore in HeLa cells (Liu et al., 2003) although we have been unable to confirm this (data not shown). It is possible that active cyclin B1-CDK1 at the inner nuclear pore potentiates MPS1 activity (Morin et al., 2012), exactly where it is needed to release MAD1 from the nuclear pore. A role for MPS1 in helping the release of MAD1 from TPR can explain why it has to be inhibited before mitosis to prevent MAD1 localization to unattached kinetochores (Hewitt et al., 2010), i.e., before release of MAD1 from the NPC.

It is intriguing to note that Plk1 and MPS1 recognize the same primary consensus motif (φ-D/E-X-S; Dou et al., 2011) and that they can cooperate by phosphorylating the same sites on proteins at the kinetochore (von Schubert et al., 2015). Therefore, it is tempting to speculate that some of the NPC components postulated to be phosphorylated by Plk1 might also be substrates of MPS1.

Once MAD1 has been released from TPR, it binds to unattached kinetochores, where it can continue to recruit cyclin B1. The binding of cyclin B1 to unattached kinetochores has been observed by a number of groups (Bentley et al., 2007; Chen et al., 2008; Alfonso-Pérez et al., 2019), and this "guilt by association" is one piece of evidence implicating cyclin B1-Cdk1 in the mechanics of the SAC and chromosome attachment. The problem in interpreting previous studies designed to elucidate the role of cyclin B1-Cdk1 in the SAC is the many SAC-independent roles that the kinase plays in mitosis: preventing cells from separating their sister chromatids or exiting mitosis; maintaining outer kinetochore structures; preventing the activation of Cdh1; and repressing phosphatase activity (Holt et al., 2008; Qian et al., 2015; Visintin et al., 1998; Zachariae et al., 1998). Additional caveats are introduced in studies using small molecule inhibitors, which can affect other cyclin-Cdk family members and related kinase families. Thus, it has been difficult to ascribe a direct role for cyclin B1-Cdk1 in the SAC. We have overcome these problems by identifying and characterizing a point mutant of MAD1 that prevents cyclin B1 from being recruited to the kinetochore but leaves the rest of the cyclin B1-Cdk1 population active in the cell. A recent study used a large deletion mutant of MAD1 to address the same question, but this mutant lacked 100 amino acids from the N terminus of MAD1 (Alfonso-Pérez et al., 2019), thereby removing a number of other important functional domains, including the nuclear localization signal that is required for it to bind to the NPC (our observation) and the ability to form a stable putative coiled-coil region that may contribute to kinetochore binding. This study concluded that by binding cyclin B1-CDK1, MAD1 increased MPS1 recruitment to kinetochores. We show here that MAD1 also has to bind cyclin B1 to be efficiently released from the NPC and properly recruited to the kinetochore.

MAD1 binding to cyclin B1 could subsequently play a role at unattached kinetochores later in mitosis; indeed, Allan et al. (2020) have recently presented evidence that cyclin B1 may form a scaffold that recruits MAD1 at the corona of prometaphase kinetochores to maintain the SAC. Nevertheless, our ability to strengthen significantly the SAC in the MAD1 mutants by ectopically targeting cyclin B1 to the NPC through POM121 shows that localized cyclin B1-Cdk1 is important for the proper control of mitosis. This targeting experiment was not perfect, however, since POM121 is at the nuclear basket, and this may be why MAD1/MAD2 was only released 2 to 4 min before NEBD and thus only partially rescued the SAC defect.

Finally, our study reveals a mechanism by which the cell uses cyclin B1-Cdk1 to coordinate activation of the APC/C at NEBD with its immediate inhibition by the MCC to ensure genomic stability. We show here that cyclin B1-Cdk1 binding to MAD1 cooperates with MPS1 to trigger MAD1/MAD2 release and recruitment to the newly formed kinetochore 10 min or more before NEBD and APC/C activation (den Elzen and Pines, 2001; Di Fiore and Pines, 2010; Geley et al., 2001). Thus, our results reveal that kinetochores recruit components necessary to generate the MCC several minutes before NEBD as the APC/C is activated (Gavet and Pines, 2010), which should be sufficient time to generate a pool of MCC to inhibit the APC/C immediately

upon NEBD. This model has the benefit that it simplifies the mechanisms required to inhibit the APC/C in early mitosis since the source of the MCC is the canonical unattached kinetochore. Its importance is underlined by the genomic instability manifested when MAD1 can no longer bind to cyclin B1.

## Materials and methods

### Plasmids and cell lines

MAD1 was tagged at the C terminus with a 3xHA-Flag epitope by PCR and cloned into a modified version of pcDNA5 FRT/TO (Thermo Fisher Scientific). Full-length MAD1 carrying either L51G/E52A/E53A/R54A/E56 or E53Q/E56Q mutations were tagged at the C terminus with mRuby by sub-cloning into the pMCSV vector. The POM121 coding region was amplified by PCR from POM121-EGFP$_3$ plasmid (kind gift from M. Hetzer, Salk Institute, La Jolla, CA) and tagged at the C terminus by sub-cloning into a modified version of pMCSV containing either mTurq2 or GBP (GFP-binding protein)-mTurq2. To generate stable cell lines, parental RPE1 and clones 7D2 and 8B12 all expressing cyclin B1-Venus:Ruby-MAD2:RFP670-MIS12 were transfected with POM121-mTurq2 and POM121-GBP-mTurq2, and cells were selected with 0.4 µg/ml neomycin (GIBCO). All constructs were verified by sequencing, and sequences are available on request.

### Cell culture and synchronization

HeLa FRT/TO cells were maintained in Advanced DMEM (Gibco) supplemented with 2% FBS, GlutaMAX (Invitrogen), penicillin (100 U/ml), streptomycin (100 µg/ml), and Fungizone (0.5 µg/ml). RPE1 cells were cultured in F12/DMEM (Sigma-Aldrich) medium supplemented with GlutaMAX (Invitrogen), 10% FBS (Gibco), 0.348% sodium bicarbonate, penicillin (100 U/ml), streptomycin (100 µg/ml), and Fungizone (0.5 µg/ml). Cells were maintained in a 37°C incubator with 5% $CO_2$. HeLa FRT/TO cells were transfected using the Flp-in-System (Thermo Fisher Scientific). Cells were induced with tetracycline (1 µg/ml; Calbiochem) 12 h before harvesting. HeLa FRT/TO cells were synchronized in S phase by a double thymidine (2.5 mM) block, then either released for 10 h for G2 phase arrested extracts, or for mitotic cells, released into nocodazole (0.33 mM) for 14 h before mitotic cells were collected by shake off. RPE1 cells were synchronized in G2 phase through a 24-h treatment with 100 nM Palbociclib (Selleckchem) followed by 14 h release into fresh medium.

### Drug treatments

For live-cell experiments, cells were treated with 50 nM sirDNA (Tebu-Bio) for 3 h before filming. AZ3146 (0.62 µM, Tocris), Taxol (100 nM, Sigma-Aldrich), reversine (166 nM, Cambridge BioScience), and nocodazole (55 nM, Sigma-Aldrich) were added just before filming.

### Genome editing

Genome editing was performed using CRISPR/Cas9D10A technology. For the MAD1 E53/56K mutation, a donor plasmid (pJ241-305516 MAD1 E53K/E56K, synthesized by ATUM)

comprising 12 silent point mutations in addition to the E53/56K substitutions (see Fig. S2, A and B) and flanked by 400 bp (5′) and 800 bp (3′) sequences, was linearized through NotI digestion. The linearized plasmid was purified (GeneJET Gel Extraction Kit, Thermo Fisher Scientific), and cotransfected into RPE1 cyclin B1-Venus:Ruby-MAD2 (Collin et al., 2013) together with a modified version of the PX466 "All-in-One" plasmid (Chiang et al., 2016) containing Cas9D10A-T2A-RFP670 and gRNAs targeting MAD1 exon4 (5′-TCACTGAGGATTCTG TTTTT-3′ and 5′-GGTGCGACCTGCTCAGCTGG-3′). RFP670-expressing cells were selected using a FACSAria III (BD Biosciences) and sorted individually into a 96-well plate. For genotyping, genomic DNA was prepared using DirectCell-PCR Lysis-Reagent Cell (VWR) according to the manufacturer's protocol and screened by PCR using a FailSafe PCR kit (Buffer E, Epicentre). The presence of MAD1 E53/56K substitutions was identified through PCR using forward primers annealing to the mutated or the wild-type sequences (5′-AGCTGGAAAAGAGGG CGAAAC-3′ and 5′-TAAGTGCCGGGAGATGCTG-3′, respectively) and the same reverse primer (5′-AGCCCACACAACGCACAC CGA-3′). Positive clones for the E53/56K mutations were screened using primers annealing ~200 bp upstream and downstream of the point mutations. PCR products were separated on agarose gels, cloned into the pDrive vector (Qiagen), and sequenced as shown in Fig. S2. The MIS12 locus was targeted with RFP670 as shown in Fig. S4. A donor plasmid containing RFP670 sequence in frame with MIS12 exon1 flanked by homology regions was cotransfected with the "All-in-One" plasmid comprising MIS12 specific gRNAs (5′-ATGACCTAC GAGGCCCAGTT-3′ and 5′-CGCCACAAACGTGCATGCTT-3′) and Cas9D10A-T2A-EGFP. EGFP-positive cells were selected via FACS then sorted individually by FACS 10 d later. RFP670 positive clones were identified by PCR (as shown in Fig. S4 B) and subsequently analyzed by live-cell microscopy to confirm MIS12 expression and localization (Fig. S4 C).

### Metaphase spreads

Cells were treated with 0.1 µg/ml colcemid (GIBCO) for 3 h, trypsinized, washed twice with 1× PBS, and resuspended in hypotonic buffer (0.075 M KCl). After a 20 min incubation at 37°C, cells were centrifuged, and the pellet was gently resuspended in 3:1 methanol/glacial acetic acid fixative (vortex dropwise). Cells were washed with fixative three times, and a few drops were released onto an alcohol-cleaned slide and allowed to air-dry. Slides were counterstained with KaryoMAX Giemsa Stain Solution (GIBCO). Transmitted light images of metaphase spreads were captured using a 63× 1.4 NA lens, and the number of chromosomes per cell was counted using ImageJ software.

### Immunoprecipitation

Cells were lysed in lysis buffer (0.5% NP-40 wt/vol, 140 mM NaCl, 10 mM KCl, 50 mM Hepes, pH 7.2, 10% wt/vol glycerol, 1 mM EDTA, and HALT protease inhibitor cocktail; Thermo Fisher Scientific). Supernatants from 11,000 x$g$ centrifugation of cell lysates were incubated with anti-Cyclin B1 (GNS1, Phar-Mingen) or anti-MAD1 (9B10, Sigma-Aldrich) antibodies

coupled to Protein G-Dynabeads (Thermo Fisher Scientific) for 1 h at 4°C, washed four times in lysis buffer, and eluted at 65°C for 5 min before analysis by SDS-PAGE and silver or Colloidal Blue staining, or immunoblotting. Silver staining was performed according to manufacturers' instructions (SilverQuest, Sigma-Aldrich). For Colloidal Blue staining, the gel was fixed in 40% methanol and 2% acetic acid for 30 min, then stained with Brilliant Blue G solution (Sigma-Aldrich), prepared overnight as per the manufacturer's instructions, then destained with 30% methanol until the background was clear.

### Mass spectrometry (MS)
For MS analyses, immunoprecipitates on Protein-G Dynabeads (Thermo Fisher Scientific) were washed 2× with triethylammonium bicarbonate buffer (100 mM) and incubated with trypsin (Roche) at 37°C for 18 h. The tryptic peptides were collected and tandem mass tag (TMT)-labeled according to manufacturers' instructions (ThermoFisher Scientific). The TMT peptides were fractionated on a U3000 HPLC system (Thermo Fisher Scientific) using an XBridge BEH C18 column (2.1 mm internal diameter [i.d.] × 15 cm, 130 Å, 3.5 µm; Waters) at pH 10, with a 30-min linear gradient from 5–35% acetonitrile/$NH_4OH$ at a flow rate at 200 µl/min. The fractions were collected every 30 s into a 96-well plate by rows, then concatenated by columns to 12 pooled fractions and dried in a SpeedVac. The peptides were redissolved in 0.5% formic acid (FA) before liquid chromotography tandem mass spectrometry (LC-MS/MS) analysis. The LC-MS/MS analysis were performed on the Orbitrap Fusion Lumos mass spectrometer coupled with U3000 RSLCnano UHPLC system (Thermo Fisher Scientific). The peptides were first loaded to a PepMap C18 trap (100 µm i.d. x 20 mm, 100 Å, 5 µm) for 8 min at 10 µl/min with 0.1% $FA/H_2O$, then separated on a PepMap C18 column (75 µm i.d. × 500 mm, 100 Å, 2 µm) at 300 nl/min and a linear gradient of 8–30.4% acetonitrile/0.1% FA in 120 min/cycle at 150 min for each fraction. The data acquisition used the SPS5-MS3 method with Top Speed at 3 s per cycle time. The full MS scans (m/z 375–1500) were acquired at 120,000 resolution at m/z 200, and the automatic gain control (AGC) was set at $4 \times 10^5$ with 50 ms maximum injection time. The most abundant multiply-charge ions (z = 2–5, above 10,000 counts) were subjected to MS/MS fragmentation by collision-induced dissociation (35% collision energy) and detected in an ion trap for peptide identification. The isolation window by quadrupole was set m/z 0.7, and AGC at 10,000 with 50 ms maximum injection time. The dynamic exclusion window was set ±7 ppm with a duration at 40 s, and only single charge status per precursor was fragmented. Following each MS/MS, the five-notch MS/MS/MS (fragment MS) was performed on the top five most abundant fragments isolated by synchronous precursor selection. The precursors were fragmented by higher-energy collisional dissociation at 65% CE, then detected in Orbitrap at m/z 100–500 with 50,000 resolution for peptide quantification data. The AGC was set at 100,000 with maximum injection time at 105 ms.

### Data analysis
The LC-MS/MS data were processed in Proteome Discoverer 2.2 (Thermo Fisher Scientific) using SequestHT and Mascot search engines against the SwissProt protein database (v. August 2018) plus the cRAP contaminant database (ftp://ftp.thegpm.org/fasta/cRAP). The precursor mass tolerance was set at 15 ppm, and the fragment ion mass tolerance was set at 0.5 D. Spectra were searched for fully tryptic peptides with maximum of two miscleavages. TMT6plex (peptide N terminus, K) was set as static modification, and dynamic modifications included deamidation (N, Q), oxidation (M), and phosphorylation (S, T, Y). Peptides were validated by Percolator with q value threshold set at 0.05 for the decoy database search. Phosphorylation site locations were verified by the *ptmRS* module. The search result was filtered to achieve a protein false discovery rate of 0.05. The TMT10plex reporter ion quantifier used 20 ppm integration tolerance on the most confident centroid peak at the MS3 level. Only unique peptides were used for quantification. The co-isolation threshold was set to 100%. Peptides with average reported S/$n$ >3 were used for protein quantification. Only master proteins were reported. Only proteins with quantification values in all samples were used for further analyses. Protein abundances were normalized to the bait protein in each immunoprecipitation subset. To filter out nonspecific proteins, a Linear Models for Microarray–based differential analysis was performed comparing MAD1 immunoprecipitations among themselves or versus IgG control samples. Proteins were deemed significantly different if adjusted P < 0.05 and twofold difference in abundance.

### Immunoblotting
For immunoblot analyses, cell lysates were loaded and run on a 4–12% NuPAGE gel (Invitrogen) and transferred to an Immobilon-FL polyvinylidene fluoride membrane (IPFL00010, Millipore) before immunoblotting (Di Fiore and Pines, 2010). Primary antibodies were used at the indicated concentrations: anti-MAD1 (clone 9B10, 1:400, mouse, Sigma-Aldrich, 2 mg/ml), anti-Flag (M2, F3165, 1:4,000, mouse, Sigma-Aldrich), anti-Mad2 (A300-301A; 1:1,000, rabbit, Bethyl Laboratories), and anti-cyclin B1 (GNSI, 1:500, PharMingen). IRDye800CW donkey anti-mouse (926–32212, LI-COR), IRDye800CW donkey anti-rabbit (926–32213, LI-COR), IRDye680CW donkey anti-mouse (926–68072, LI-COR), and IRDye680CW donkey anti-rabbit (926–68073, LI-COR) secondary antibodies were all used at 1:10,000. Quantitative immunoblotting was performed on a LI-COR Odyssey CLx scanner according to the manufacturer's instructions (LI-COR Biosciences).

### Immunofluorescence
To costain for MAD1 and TPR, cells were fixed for 15 min at room temperature in 60 mM Pipes, 25 mM Hepes, 10 mM EGTA, 2 mM $MgCl_2$, pH 6.9, buffered with KOH) buffer with 4% wt/vol paraformaldehyde and 0.5% vol/vol Triton X-100 . Primary antibodies were diluted as follows: TPR (HPA024336, rabbit, Atlas antibodies) 1:50; MAD1 (9B10, Sigma-Aldrich, 2 mg/ml) 1:200 and human anti--centromere autoantibody (ACA) (CS1058, human, Cortex Biochem) 1:200. To costain for cyclin B1 and TPR, cells were fixed for 2 min in 50% methanol and 50% acetone and washed with PBS. Primary antibodies were diluted as follows: anti-cyclin B1 (GNSI, PharMingen) 1:100, and anti-TPR as above.

Secondary antibodies were anti-mouse-594 nm, anti-rabbit-488 nm, and anti-human-647 nm (Alexa Fluor, Thermo Fisher Scientific), all at 1:400. Confocal imaging of antibody-stained samples was performed on a Marianas microscope (Intelligent Imaging Innovations).

### Colocalization analysis

To measure the colocalization of TPR and MAD1 a 4.2 μm² representative region of interest (ROI) was defined from the maximum projection image of each cell. The percentage TPR and MAD1 colocalization was calculated with the Fiji plugin "Colocalization Threshold" (National Institutes of Health, public domain; colocalization threshold plug-in: authors Tony Collins and Wayne Rasband).

To measure the colocalization of TPR and cyclin B1 at the nuclear membrane, cells were costained with anti-TPR and anti-cyclin B1 antibodies and visualized using a Super-resolution via optical reassignment (SoRa) Spinning Disc microscope (Intelligent Imaging Innovations and Yokogawa). Late prophase cells just before NEBD were identified by the localization of cyclin B1 and the extent of chromosome condensation. From each confocal Z-section image taken through the center of each cell, a representative 6-μm-long ROI of TPR signal at the nuclear membrane was manually defined. Distribution profiles of the immunofluorescence intensities of TPR and cyclin B1 along each ROI were measured with Fiji software. The percentage of TPR and cyclin B1 colocalization was calculated as the area of the TPR immunofluorescence intensity profile that overlapped with that of cyclin B1 (OriginLab2020).

### Time-lapse imaging and analysis

For time-lapse microscopy, cells were seeded and transfected on an eight-well chamber slide (μslide, Ibidi). Cells were pretreated with 50 nM Sir-DNA (Spirochrome) 3 h before filming to visualize chromosomes. Cells were imaged in Leibovitz's L-15 medium (GIBCO) supplemented with 10% FBS. Time-lapse confocal imaging was performed on a Marianas confocal spinning-disk microscope system (Intelligent Imaging Innovations, Inc.) comprising a laser stack for 445 nm/488 nm/514 nm/561 nm lasers; an Observer Z1 inverted microscope (Carl Zeiss) equipped with Plan-Apochromat 40× 1.3 NA and 63× 1.4 NA lenses; an OKO stage top incubator set to 37°C (OKO); a CSU X1 spinning disk head (Yokogawa); a Gemini W view optical splitter attached to a Flash4 CMOS camera (Hamamatsu), and a QuantEM 512SC camera (Photometrics). The microscope was equipped with Brightline filters (Semrock) for GFP/RFP, for CFP/YFP/RFP, and for RFP670. Immunofluorescence images were captured on a similar Marianas confocal microscope but equipped with a CSU W1 head. Colocalization of Ruby-MAD2 with RFP670-MIS12 and immunofluorescence images were collected using a 63× 1.2 NA objective (Carl Zeiss). Time-lapse widefield fluorescence and differential interference contrast (DIC) imaging were performed on a Nikon Eclipse microscope (Nikon) equipped with 20× 0.75 NA, 40× 1.3 NA, and 63× 1.4 NA lenses, a Flash 4.0 CMOS camera (Hamamatsu), an excitation and an emission filter wheel equipped with Brightline (Semrock) filters for CFP, GFP, YFP, RFP, and RFP670, and an analyzer in the emission wheel for DIC

imaging. Image acquisition and processing for the confocal microscopes was performed using Slidebook 6 (Intelligent Imaging Innovation, Inc.) software; Micromanager software and ImageJ open source software were used for widefield imaging.

3D videos of Ruby-MAD2 localization with RFP670-MIS12 of single cells were quantified using an open source program (https://github.com/adamltyson/coloc-3DT). Images were converted to Open Microscopy Environment-Tagged Image File Format (OME-TIFF), loaded into a custom python program, resliced in Z to isotropic sampling, and smoothed with a Gaussian filter (sigma = 1 voxel). To segment the kinetochores, the RFP670-MIS12 signal was thresholded using an adaptation of Otsu's method (Otsu, 1979) in which the threshold was scaled by a fixed value (1.08) for all experiments. Noise was removed by morphological opening (kernel = 1 voxel cube), and then the mean value (colocalization) of MAD2-Ruby was calculated within the thresholded kinetochores. This colocalization was scaled to the level of Ruby-MAD2 within the rest of the nucleus (estimated as between 1 and 15 voxels from the segmented kinetochore). All image processing was performed with Scikit-image (van der Walt et al., 2014).

### Statistics

Statistical analyses were performed using GraphPad Prism. Significance of data derived from mitotic timings was determined using unpaired Mann–Whitney tests. Statistical analyses of MS were performed using the R package LIMMA, and proteins with a logFC >2-fold and adjusted P value <0.05 were considered significant. Binding to cyclin B1 was analyzed by unpaired Student's $t$ tests. All P values are two-tailed. Data distribution was assumed to be normal, but this was not formally tested.

### Online supplemental material

Fig. S1 provides supporting evidence that cyclin B1 binds to a helical surface on MAD1. Fig. S2 shows the design of the guide RNAs to mutate MAD1 and the genomic sequence of the mutated MAD1 clones. Fig. S3 shows the mitotic delay of the MAD1 wild-type and mutant clones induced by Taxol treatment. Fig. S4 shows the effect of the AZ3146 MPS1 inhibitor on the timing of NEBD to anaphase in parental and MAD1 mutant clones. Fig. S5 shows evidence supporting the targeting of RFP670 into the Mis12 locus to label kinetochores. Videos 1, 2, and 3 are time-lapse videos of fluorescently tagged cyclin B1 and MAD2 as wild-type (Video 1) and two MAD1 E53/56K mutant clones (Videos 2 and 3) enter mitosis. Videos 4, 5, 6, and 7 are time-lapse videos showing the kinetochore recruitment of cyclin B1 and MAD2 in parental (Videos 4 and 6) and a MAD1 E53/56K mutant clone (Videos 5 and 7) upon treatment with an MPS1 inhibitor. Videos 8 and 9 show the kinetochore recruitment of MAD2 in parental (Video 8) and a MAD1 E53/56K mutant clone (Video 9) when cyclin B1 is recruited to the NPC by Pom121.

### Acknowledgments

We thank Will Chiang and Steve Jackson (Gurdon Institute, Cambridge, UK) for the CRISPR-Cas9[D10A] construct; Martin

Hetzer (Salk Institute, La Jolla, CA) for POM121 cDNA; Marco Chiapello for help with LIMMA analysis; Andy Riddell, Fredrik Wahlberg, and Rhadika Patel for help with fluorescence-activated cell sorting; Oxana Nashchekina for help with designing the strategy to mutate MAD1 in RPE1 cells; Federica Schiavoni for advice on metaphase spreads; Iain Hagan and Eleanor Trotter for the original idea and advice on synchronizing cells with Palbociclib; and all the Cell Division Team for helpful discussions.

This work was supported by a CR UK Program grant to J. Pines and by Institute of Cancer Research (ICR) funds to support C. Marcozzi.

The authors declare no competing financial interests.

Author contributions: M. Jackman identified MAD1 binding to cyclin B1 the E53/56 MAD1 residues necessary for binding to cyclin B1. M. Jackman and C. Marcozzi generated and characterized the MAD1 E53/56K mutants. M. Jackman, C. Marcozzi, M. Pardo, L. Yu, and J.S. Choudhary performed the mass spectrometry. C. Marcozzi and M. Pardo analyzed the data. A.L. Tyson and M. Jackman designed and wrote the coloc-3DT program. M. Jackman and M. Barbiero performed and analyzed the TPR-cyclin B1, TPR-MAD1, and MAD2-MIS12 colocalization experiments and quantified cyclin B1-Venus degradation. J. Pines and C. Marcozzi designed and executed the POM121 rescue experiment. M. Barbiero, M. Jackman, C. Marcozzi, and J. Pines analyzed the data and wrote the paper.

Submitted: 12 July 2019

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

# Supplemental material

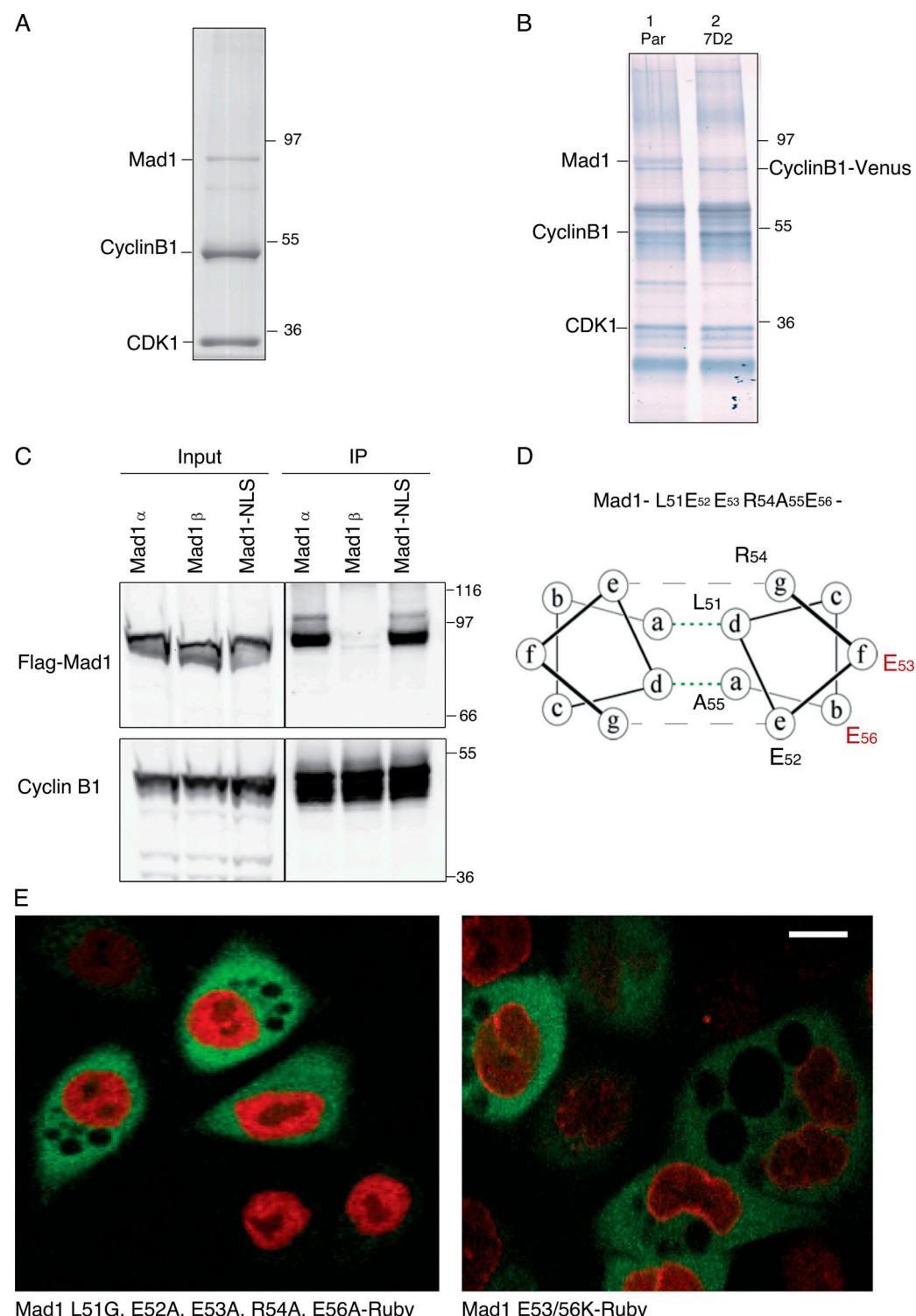

Figure S1. **Cyclin B1 binds to MAD1 through the acidic face of a helix encoded by exon 4.** Related to Fig. 1. **(A)** Colloidal blue stained SDS-PAGE gel of cyclin B1 immunoprecipitated from HeLa cells. Marked bands were excised and identified by mass spectrometry. **(B)** Silver-stained SDS-PAGE gel of cyclin B1 immunoprecipitates from RPE cyclin B1-Venus$^{+/-}$: Ruby-MAD2$^{+/-}$ cells (lane 1) and MAD1 E53K/E56K clone 7D2 (lane 2). **(C)** Cyclin B1 immunoprecipitates from HeLa cells expressing Flag-epitope tagged MAD1α or MAD1β or MAD1 with a mutated nuclear localisation sequence KKR79-82AAA (Mad1-NLS), probed with anti-FLAG (upper panel) or anti-cyclin B1 (lower panel) antibodies. **(D)** Heptad registration of acidic residues of MAD1 within coiled-coil configuration, predicted using PairCoil2. **(E)** Confocal image of HeLa cyclin B1-Venus$^{+/-}$ (green) cells transfected with either MAD1 L51G/E52A/E53A/R54A/E56A-Ruby (left panel, red) or MAD1 E53/56K-Ruby (right panel, red). Scale bar, 10 μm.

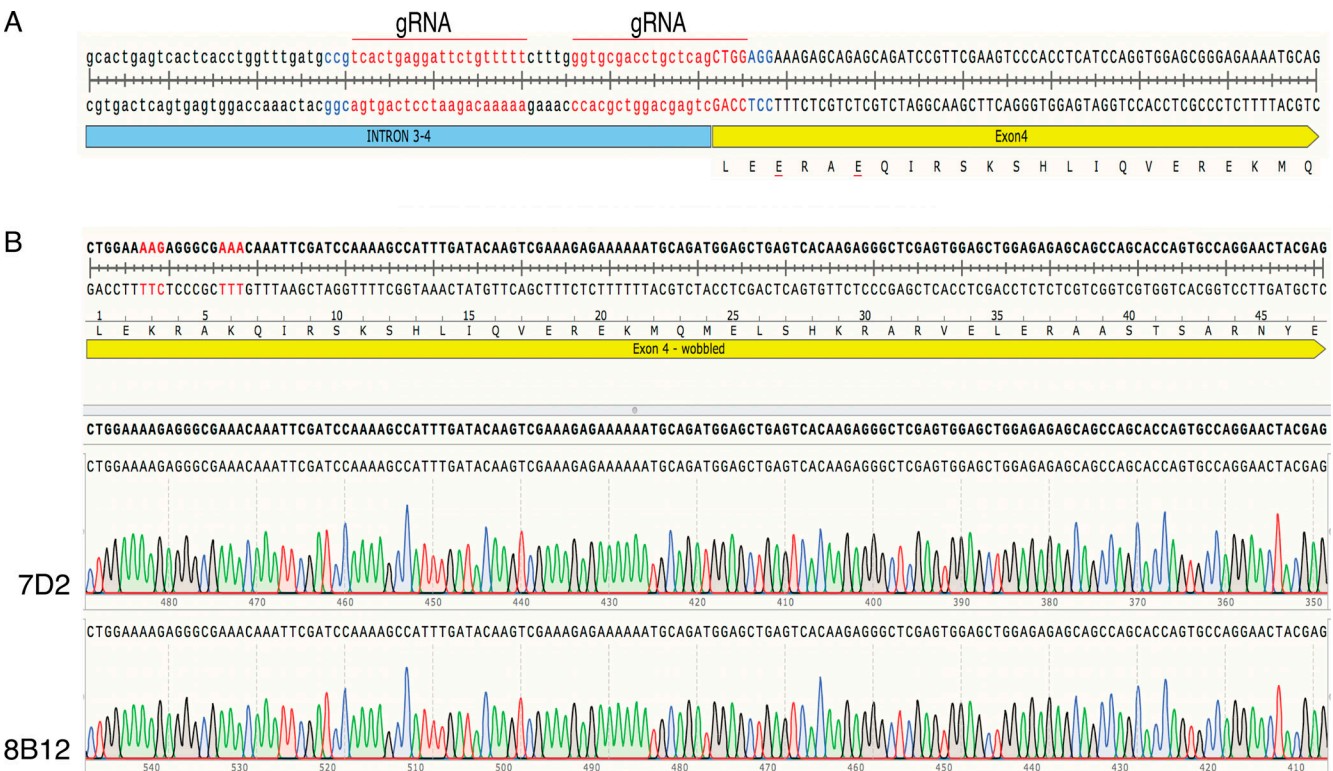

Figure S2. **MAD1 recruits cyclin B1 to kinetochores.** Related to Fig. 2. **(A)** Schematic showing guide RNA (gRNA, red) selection and protospacer adjacent motif (blue) for CRISPR-Cas9$^{D10A}$ targeting of MAD1 exon4. **(B)** Genomic DNA sequencing of RPE cyclin B1-Venus$^{+/−}$:MAD2-Ruby$^{+/−}$ MAD1 E53/56K clones 7D2 and 8B12.

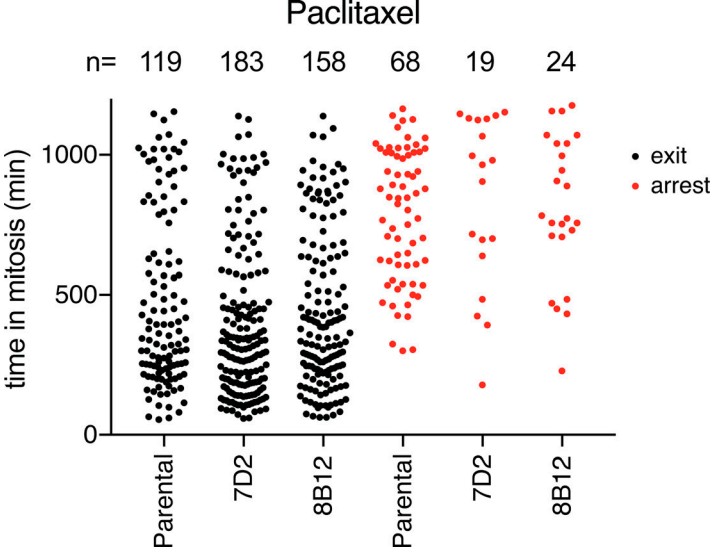

Figure S3. **MAD1 binding to cyclin B1 is required for genomic stability**. Related to Fig. 3. The duration of the mitotic arrest for parental RPE cyclin B1-Venus$^{+/−}$:Ruby-MAD2$^{+/−}$ cells and the MAD1 E53/E56K clones 7D2 and 8B12 was assayed by time-lapse DIC microscopy in 100 nM paclitaxel and plotted as black dots. The times in mitosis of cells that remained arrested for the duration of the experiment are shown separately as red dots. Total number of cells for each clone analyzed; parental $n$ = 187 cells, clone 7D2 $n$ = 201 cells, clone 8B12 $n$ = 182 cells.

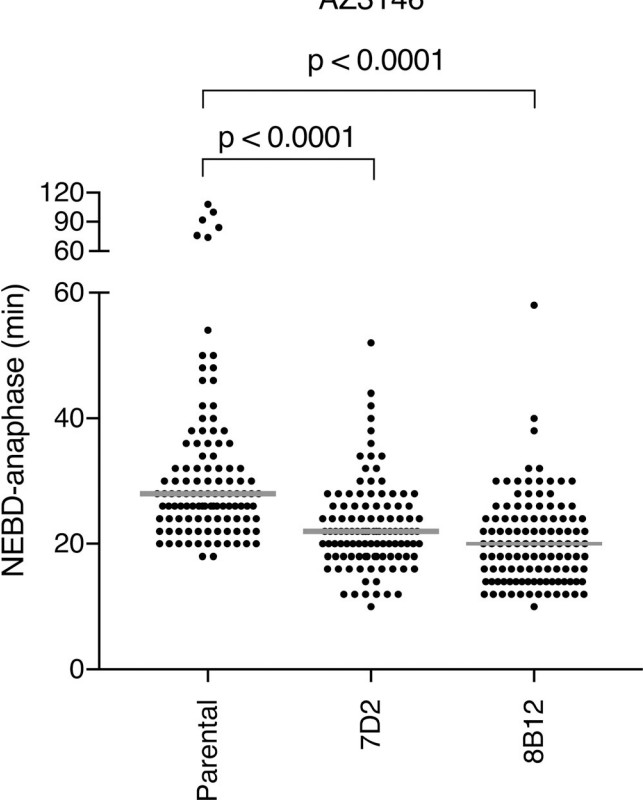

Figure S4.   **Cells with MAD1 mutants that cannot bind cyclin B1 are sensitive to partial inhibition of MPS1.** Related to Fig. 4. **(A)** Time from NEBD-anaphase for parental RPE cyclin B1-Venus$^{+/-}$:MAD2-Ruby$^{+/-}$ cells and MAD1 E53/56K clones 7D2 and 8B12 treated with 0.62 µM AZ3146 MPS1 kinase inhibitor. Scatter dot pots show the median (gray line) from two independent experiments (parental $n$ = 112 cells, 7D2 $n$ = 111 cells, 8B12 $n$ = 117 cells). The two-way P values were calculated using a Mann–Whitney unpaired $t$ test.

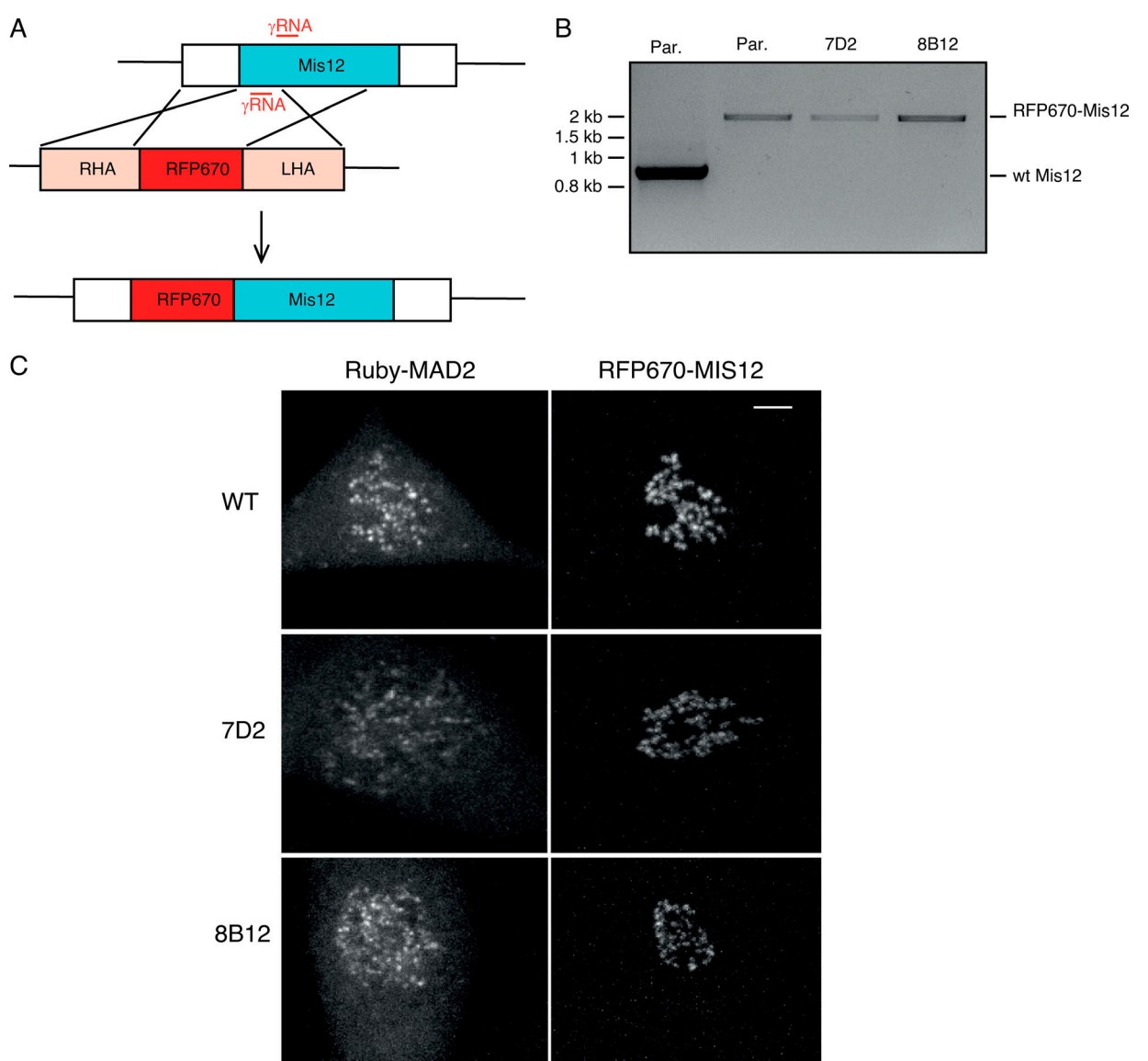

Figure S5.   **MAD2 recruitment to kinetochores is delayed when MAD1 cannot bind cyclin B1.** Related to Fig. 5. **(A)** Schematic showing how RFP670 was tagged at the N terminus of MIS12 (RHA and LHA refer to right and left homology arms, respectively). **(B)** PCR of genomic DNA from wild-type RPE1 cells (control), parental RPE cyclin B1-Venus$^{+/−}$: Ruby-MAD2$^{+/−}$ RFP670-MIS12+/+ cells (Par.), or MAD1 E53/E56K: RFP670-MIS12$^{+/+}$ clones 7D2 and 8B12, showing integration of RFP670 into both alleles of MIS12. **(C)** Maximum projection images of parental RPE cyclin B1-Venus$^{+/−}$:Ruby-MAD2$^{+/−}$; RFP670-MIS12$^{+/+}$cells (Par.) and MAD1 E53/E56K: RFP670-MIS12$^{+/+}$ clones 7D2 and 8B12. Left panels show Ruby-MAD2; right panels show RFP670-MIS12. Scale bar, top right panel, 5 µm.

Video 1.   **Mitotic entry of parental RPE cyclin B1-Venus$^{+/−}$:MAD2-Ruby$^{+/−}$ cell.** Cyclin B1-Venus (left panel), Ruby-MAD2 (middle panel), and merged channels cyclin B1-Venus (green) and Ruby-MAD2 (red; right panel). Cells imaged by spinning disk confocal microscopy. Frame rate, one image every 2 min.

Video 2.   **Mitotic entry of MAD1 E53/E56K clone 7D2.** Cyclin B1-Venus (left panel), Ruby-MAD2 (middle panel), and merged channels cyclin B1-Venus (green) and Ruby-MAD2 (red; right panel). Cells imaged by spinning disk confocal microscopy. Frame rate, one image every 2 min.

Video 3.   **Mitotic entry of MAD1 E53/E56K clone 8B12.** Cyclin B1-Venus (left panel), Ruby-MAD2 (middle panel), and merged channels cyclin B1-Venus (green) and Ruby-MAD2 (red; right panel). Cells imaged by spinning disk confocal microscopy. Frame rate, one image every 2 min.

Video 4.   **Widefield epifluorescence video shows mitotic entry of parental RPE cyclin B1-Venus$^{+/-}$:Ruby-MAD2$^{+/-}$ cells treated with SiR-DNA 3 h before filming and reversine (166 nM) just before imaging.** Frame rate, one image every 2 min.

Video 5.   **Widefield epifluorescence video shows mitotic entry of RPE Cyclin B1-Venus$^{+/-}$:Ruby-MAD2$^{+/-}$MAD1E53/E56K clone 8B12 treated with SiR-DNA 3 h before filming and reversine (166 nM) just before imaging.** Frame rate, one image every 2 min.

Video 6.   **Spinning disk confocal video shows mitotic entry of parental RPE cyclin B1-Venus$^{+/-}$:Ruby-MAD2$^{+/-}$; RFP670-MIS12$^{+/+}$ cells.** Cyclin B1-Venus (far left panel), Ruby-MAD2 (left panel, green), RFP670-MIS12 (right panel, red), and merged channels for MAD2 and MIS12 (far right panel). Cells treated with reversine (166 nM) just before imaging. Note that reversine reduces the loading of cyclin B1 onto kinetochores in the parental cells. Frame rate, one image every 2 min.

Video 7.   **Spinning disk confocal video shows mitotic entry of MAD1 E53/E56K clones 7D2.** Cyclin B1-Venus (far left panel), Ruby-MAD2 (left panel, green), RFP670-MIS12 (right panel, red), and merged channels for MAD2 and MIS12 (far right panel). Cells treated with reversine (166 nM) just before imaging. Frame rate, one image every 2 min.

Video 8.   **Projected Z-series stacks of cells filmed using confocal spin disk microscopy with 2-min time frame.** Parental RPE cyclin B1-Venus$^{+/-}$:Ruby-MAD2$^{+/-}$:RFP670-MIS12$^{+/+}$ expressing POM121-GBP-mTurq2. Panels, left to right, show cyclin B1-YFP, Mad2-Ruby, RFP670-Mis12, and then merged image. Merged image Mad2-Ruby (green), RFP670-Mis12 (red).

Video 9.   **Projected Z-series stacks of cells filmed using confocal spin disk microscopy with 2-min time frame.** MAD1 E53/E56K clone 7D2 expressing POM121-GBP-mTurq2. Panels, left to right, show cyclin B1-YFP, Mad2-Ruby, RFP670-Mis12, and then merged image. Merged image Mad2-Ruby (green), RFP670-Mis12 (red).

