## [Peer Review File · The Journal of Cell Biology]

Cyclin B1-Cdk1 facilitates MAD1 release from the nuclear pore to ensure a robust Spindle Checkpoint

Mark Jackman, Chiara Marozzi, Martina Barbiero, Mercedes Pardo, Lu Yu, Adam Tyson, Jyoti Choudhary, and Jonathon Pines

Corresponding Author(s): Jonathon Pines, Institute of Cancer Research

Review Timeline:

Submission Date:	2019-07-12
Editorial Decision:	2019-08-01
Revision Received:	2020-02-05
Editorial Decision:	2020-02-27
Revision Received:	2020-03-03

Monitoring Editor: Arshad Desai

Scientific Editor: Tim Spencer

Transaction Report:

DOI: <https://doi.org/10.1083/jcb.201907082>

August 1, 2019

Re: JCB manuscript #201907082

Prof. Jonathon Pines
Institute of Cancer Research
Cancer Biology
237 Fulham Road
London SW3 6JB
United Kingdom

Dear Jon,

Thank you for submitting your manuscript entitled "Cyclin B1-Cdk1 binding to MAD1 links nuclear pore disassembly to chromosomal stability" to JCB. The manuscript has been evaluated by expert reviewers, whose reports are appended below. Unfortunately, after an assessment of the reviewer feedback, our editorial decision is against publication in JCB.

You will see that the reviewers provide a mixed evaluation of the study. Reviewers #1 and #2 both raise substantial concerns about the major conclusion presented in the manuscript and consequently do not give the work high priority; Reviewer #2 is additionally concerned about overlap with the Barr manuscript published earlier this year. Reviewer #3 is more supportive of the study but raises specific concerns related to a subset of the experiments.

We have discussed the reviewer reports and also considered the need for expediency given the published work and the additional study under consideration elsewhere. While we are willing to waive concern about overlap with the Barr manuscript, a more serious concern is that two expert reviewers raise substantial criticisms about the data underlying the major conclusion presented in your manuscript. These concerns, unfortunately, make it not possible for us to further consider the submitted manuscript. We believe that addressing the significant reviewer comments to provide stronger support for the model presented will require substantial experimental investment and may work against the need for expediency. We are therefore returning the manuscript to you with the hope that the comments will help you improve it for publication elsewhere in a timely manner. If you feel that you are able to address the major concerns related to the central conclusion of the work and persuade Reviewers #1 & #2 about the validity of your model, you may consider an appeal and re-submission. However, any such re-submission will need to be evaluated at the time it is received for priority and novelty as per JCB policy.

We regret that our decision could not be more positive, but hope that you find the reviews constructive. Of course, this decision does not imply any lack of interest in your work and we look forward to future submissions from your lab.

Thank you for your interest in the Journal of Cell Biology.

Sincerely,

Arshad Desai, PhD

Editor, Journal of Cell Biology

Melina Casadio, PhD
Senior Scientific Editor, Journal of Cell Biology

Reviewer #1 (Comments to the Authors (Required)):

Summary:

Jackman and colleagues here investigate the role of a CyclinB1/Cdk1-Mad1 physical interaction in the regulation of spindle assembly checkpoint (SAC) signaling in cultured human cells. This is an area of intense interest, as Mad1, a key signaling component of kinetochore-mediated checkpoint signaling during mitosis, is localized to the nuclear pores during interphase and how Mad1 is released from the pores is an unresolved question. Recently, the Barr lab has shown that Mad1 recruits CyclinB1/Cdk1 to unattached kinetochores in human cells and that this recruitment is necessary for robust SAC signaling (Alfonso-Pérez et al., 2019 JCB). Extending upon these findings, Jackman and colleagues identify key residues mediating the Mad1/CyclinB1 interaction. They go on to perform a number of microscopy assays to argue that loss of this interaction (through CRISPR generated mutations) leads to defects in checkpoint signaling, and critically, inability of Mad1 to disassociate from nuclear pores during mitosis. Based on these data, the authors propose that CyclinB1/ Cdk1 activity coordinates Mad1 release from the nuclear pores prior to nuclear envelope breakdown to regulate the SAC and maintain genomic stability.

These observations, if reproducible and robust, would certainly be of great interest to the field as they move it closer to a mechanistic understanding of how Mad1 localization is regulated to mediate the SAC. However, there are experimental and textual concerns that must be addressed before reevaluating whether or not CyclinB1 interactions regulate Mad1 localization and checkpoint response in human cells.

Major Concerns:

1) A key experimental concern is in the generation of clonal mutants. The authors use Mad1 mutant clones 7D2 and 8B12 throughout the manuscript, but it is troubling that the control parental line was not simultaneously single cell cloned. This brings up the possibility that many of the phenotypes observed in the mutant clones aren't necessarily the result of the E53/56K mutations but are rather due to the selection process of generating a single cell clone.

A further concern is in the heterogeneity of the clonal phenotypes. The data in figure 3 clearly show that mutant 8B12 completes unperturbed mitosis faster than the parent cell line but is not affected by nocodazole or paclitaxel treatment. This is in contrast to mutant 7D2, which has the same mitotic time as the parent but is sensitive to paclitaxel. Likewise, according to figure panels 4D and 4E, these clones have different levels of CyclinB1. Is it possible that these different levels of CyclinB1 in the clones is what is driving the observed phenotypes? Are these the only 2 clones that survived genome editing? If they are the only surviving clones then the observed mitotic phenotypes are less meaningful to us.

These concerns can be addressed experimentally by repeating the experiments entailed in Figures 3 and 4 using several clonal parental lines.

2) Figure 2A and B make it appear as though there is less overall Mad1 in the mutant clones.

Couldn't lower levels of Mad1, and not the mutations therein, explain the slower recruitment of Mad2 to kinetochores seen in Figure 5? This should be addressed experimentally or textually.

3) As mentioned in point 1 above, the phenotypes assayed in figure 3 are not consistent across clones. Due to this, the authors should tone down the claim that 'Mad1 binding to CCNB1 is required for genomic stability' or provide additional sufficient evidence.

4) Fusion of CyclinB1 to pores in mutant Mad1 background does not rescue SAC response to WT levels. This suggests that the main role for Cyclin B1 interaction with Mad1 is not at the pore but is rather perhaps at the kinetochore, as suggested by Alfonso-Pérez et al. This should be tested by fusing CyclinB1 to the kinetochore in both clones. Given the already mentioned phenotypic clonal heterogeneity, it is concerning that the current manuscript only presents the pore-fusion experiment in 1 clone.

This experiment should also examine the localization of Mad1 to determine whether that is also partially restored. The authors conclude that the partial SAC rescue is due to restoring the release of Mad1 but never assay its localization.

5) Throughout the main text, the authors use the observed phenotypes to argue that CyclinB1-Cdk1 activity is important for releasing Mad1 from the NPC. There is no data to formally support this statement (e.g. no use of Cdk1 inhibitors that phenocopy the E53/56K mutants). Please revise these statements throughout the text or show that it is indeed Cdk1 activity.

Minor Points

1. A summary of MS data should be included in the manuscript in some form.

2. The title is an overstatement. While Cyclin B/Mad1 binding regulates Mad1 release from the NPC as well as genomic stability, there is no clear evidence that they are linked. The tethering experiment never addressed chromosomal stability so this title should be a better description of the findings.

3. The Mad1/TPR colocalization in figure 6 should be quantified.

4. The introduction was a bit lengthy about issues that weren't necessarily related to the findings and could be shortened.

Reviewer #2 (Comments to the Authors (Required)):

In this paper, the authors identify a Mad1-Cyclin B interaction and investigate the behavior of a Mad1 mutant that fails to bind to Cyclin B1. This mutant Mad1 still localizes to the nuclear envelope and kinetochores, but fails to recruit Cyclin B1 to kinetochores. In addition, this Mad1 mutant mislocalizes to chromosomes outside of kinetochores and appears to colocalize with TPR, a component of the nuclear pore complex (NPC). The mutant cells also display increased genome instability and a defect in maintaining the SAC, particularly when combined with an Mps1 inhibitor, which may be due to a delay in Mad2 recruitment to kinetochores. The authors propose a model in which Mad1 binding to Cyclin B1-CDK1 promotes Mad1 release from NPC before NEBD such that there is a pool of Mad1/Mad2 poised to be recruited to kinetochores. Overall, this is an interesting model, but much more data is needed to support it. In addition, recent work from the Barr lab

published in the JCB (Alfonso-Pérez et al. 2019) conducted very similar studies on the Mad1-Cyclin B interaction and its role in checkpoint control. Based on both of these, I cannot recommend publication of this paper in the JCB.

Major comments

1. The presence of the Mad1-Cyclin B1 interaction is well established by the work in this paper and the recent paper from the Barr lab (Alfonso-Perez et al. 2019). However, whether this interaction is required at kinetochores, the nuclear envelope, in the cytoplasm, or all of the above remains unclear. For this paper, the authors make a strong case that this interaction is required at the nuclear envelope, but I found this to be the least convincing part of the paper. First, the authors state that Cyclin B1-CDK1 activity at the nuclear envelope promotes Mad1 release from the NPC and allows proper recruitment to kinetochores. However, Cyclin B1 does not show strong enrichment at the nuclear envelope before NEBD (see Figure 7A). Mad2 has also been shown to directly bind TPR, but Mad2 release from NPC appears to be independent of Cyclin B1-CDK1. Second, the most compelling experiment for this point is the attempt to rescue this through the artificial tethering of Cyclin B1-CDK1 to the nuclear envelope through POM121. However, the data for this "rescue" in Figure 7B seems quite weak. There is only a minor increase in mitotic duration in this construct, and it certainly does not approach the levels that occur in the parental cells.
2. The overlap with the recent work published by the Barr lab is a consideration, as many aspects of these papers are similar (the Barr lab identified a similar interaction domain and tested the role of this interaction in checkpoint function, etc). Ultimately, this is an editorial decision, but the data in this paper is not that different. Here, the authors have tried to emphasize different roles for this interaction and to develop a distinct model with function at the nuclear envelope, but as indicated above this is the least compelling aspect of the paper.
3. In the discussion, the authors state that "Cyclin B1-CDK1 coordinates with Mps1 kinase" to promote release of Mad1 from the NPC. Has Mps1 been shown to localize to the nuclear envelope? Alternatively, the effect of Mps1 inhibition may be independent of Mad1 release from the NPC. Instead, Mps1-mediated phosphorylation is simply required to promote Mad1 localization to kinetochores (as is the case in yeast or as was proposed in the Barr lab paper). In fact, a low dose of reversine alone leads to a reduction in Mad1 localization to kinetochores in the absence of the Mad1 localization around the chromosomes that is observed with the mutant.
4. The authors assess the timing of Mad2 localization and see a delay in recruitment to kinetochores. What about timing of Mad1 recruitment to kinetochores? Maybe the mutant Mad1 is less efficient at recruiting Mad2, which would also result in delayed Mad2 recruitment?

Minor comments

1. The data in Fig 3B-D is not very convincing to established that the Mad1 mutants have a weakened SAC. Although the authors state that there statistical differences, the plots look very similar, just more spread out. What about trying higher levels of nocodazole and/or taxol? These should cause a more potent arrest, thus making it easier to assess whether some cells fail to maintain the SAC. However, in this case, the SAC is fully active which may be a problem if the effect of the mutation is mild.
2. Fig 1A - In the IP for Mad1, very little Cyclin B1 is pulled down when compared to the amount of Mad1 observed with IP for Cyclin B1. The authors should comment on this point and explain this discrepancy.
3. The ability to generate the Mad1 mutants at the endogenous locus is powerful and a strength of this paper. However, based on the way that these were generated, these cell lines have undergone multiple divisions in the absence of the wild type protein leading to an accumulation of defects. The different behavior of the clones is notable in this regard. In an ideal world, the authors would compare this is a conditional strategy to replace Mad1. In the absence of this, they should at least

comment on this caveat and consideration.

4. Based on previous proteome-wide studies (curated at phosphosite.org), Mad1 is phosphorylated at two sites that display a CDK-like consensus (residues 423 and 428). One possibility is that this interaction serves to recruit Cyclin B- Cdk1 to Mad1 to aid in its phosphorylation. It would be helpful to comment on this point, or possibly test the consequences of eliminating CDK sites in Mad1.
5. Fig S1C - The authors state that Mad1a and Mad1b from cDNAs migrated at the same molecular masses as the two forms in cells. However, the blot is not very convincing. They could repeat this experiment to generate a better gel, or they could simply remove this data and this point as it is clearly established based on their mutant data. It should be enough to just say that Mad1a is the only one that binds Cyclin B1.
6. Figure 1C and S1E have other Mad1 mutants that are not mentioned in the text, e.g. the 5AA substitution mutant that disrupts binding and nuclear envelope localization. The authors should write a few lines in the text describing the rationale and behavior of these mutants
7. Fig 1 - the authors should mention in legend that both Cyclin B1 and Mad1 were IP'ed.
8. Fig 1C - please indicate on the figure that the different Mad1 constructs are expressed from a tetON-inducible promoter.
9. Fig 2A-B - Why is there a shift in the Mad1 band between input and after IP, especially since this shift does not occur with Mad2? Also why are the two isoforms of Mad1 not detectable in the input?
10. Fig 2C - although the strongest effect is on Cyclin B1 localization to kinetochores, the Mad2 signal is also reduced with more diffuse signal in the cytoplasm.
11. Fig 4D-E - I am confused when the authors state that the Cyclin B1-Venus is degraded with similar kinetics in wild-type and mutant cells. This may be the case in the 7D2 mutant, but the slope is different in the 8B12 mutant.
12. Fig 5A - do the cells displayed in the figure also express MIS12-RFP670? If yes, please indicate in the legend.
13. Fig 6 - It is nice to see colocalization between Mad1 and TPR around chromosomes, but if indeed the mutant remains bound to TPR, the authors should be able to observe that by immunoprecipitation.

Reviewer #3 (Comments to the Authors (Required)):

The manuscript by Pines and colleagues shows how Cyclin B1 binding to Mad1 through an acidic patch on Mad1 regulates its dissociation from Tpr at the nuclear pore, required for efficient recruitment of Mad1/Mad2 to unattached kinetochores, and consequently normal SAC response, before complete nuclear envelope breakdown. The localization of Cyclin B1 to multiple mitotic structures, including the nuclear envelope and kinetochores has been known for years, but the relevance of its localization for function has remained unclear. More recently, work by the Barr lab has provided evidence for a functional interaction between Cyclin B1 and Mad1 required for SAC signaling. The present work by the Pines lab extends beyond current knowledge as it provides a view of how Cyclin B1-Mad1 interaction, by defining the precise interaction motif on Mad1, is required to mount a SAC response while inhibiting APC at mitotic entry. Importantly, the results provided in the current manuscript clarify previous controversy regarding the role of nuclear pores in generating an MCC (Rodriguez-Bravo et al., 2014). Overall, I find this an elegant study and an important contribution that will be of interest to the JCB readership. I have only minor requests/suggestions that would like to see addressed prior to publication:

- 1- The introduction of this manuscript is an art piece as it easily, yet comprehensively, summarizes

years of literature in a rather mature field, while clearly isolating the question being addressed: how disassembly of interphase structures are coordinated with the assembly of mitosis-specific structures, relevant for SAC signaling. I have only two suggestions: first, the authors refer to the role of Cyclin B-Cdk1 in the phosphorylation of structural components, including microtubule motors, but they do not mention non-motor MAPs, which probably account for the more dynamic behavior of microtubules as cells enter mitosis; second, in parallel to the work by the Jalepalli lab (Rodriguez-Bravo et al., 2014), it would be fair to cite also the previous works by the Maiato lab on Tpr and SAC regulation (Lince-Faria et al., 2009 and Schweizer et al., 2013), which offered an alternative model of how Tpr at nuclear pores regulates subsequent kinetochore-dependent MCC assembly and SAC response, in a way very consistent with the model being now proposed by the Pines lab. In fact, the original interaction between Mad1 and Cyclin B1-Cdk1 was reported (yet not functionally explored) by Schweizer et al., 2013, precisely using a similar mass-spec approach, but pulling from Mad1, which nicely complements the strategy used in this and previous works (Barr lab) in which Mad1 was identified after pulling from Cyclin B1.

2- The authors use low doses of nocodazole to infer SAC function in the Mad1 E53/56K mutant, but they reported no effect on SAC response. Arguably, low nocodazole is known to stabilize microtubules (M.A. Jordan, D. Thrower, L. Wilson *Journal of Cell Science* 1992 102: 401-416) and the real test to SAC response would be incubation of the WT vs mutant cells in high (micromolar) doses of nocodazole. This would also test about the issue of SAC maintenance.

3- Related to the previous point, the authors argue through elegant live-cell experiments that Mad2 recruitment was delayed in the Mad1 mutant cells and this was responsible for the compromised SAC response. An experiment that would further test this hypothesis would be to delay mitosis in the mutant by a temporary inhibition of the proteasome or APC activity, say for about 10 min, and then washout the inhibitor(s) and see if now Mad2 is recruited to normal levels and infer the respective SAC response and segregation fidelity.

4- The authors refer to the role of Mps1 in Mad1 localization at unattached kinetochores only before Mad1 release from the NPC and cite prior works on Mps1 inhibition before or after mitotic entry (Hewitt et al., 2010). However, other works have shown that Mad1 recruitment and maintenance both depend on Mps1 activity (e.g. Schweizer et al., *JCB* 2013). The authors should therefore use caution here, as the relationship with Mps1 might not be so straightforward.

5- The recruitment of Mad1 to kinetochores has been a matter of intense investigation recently. The results provided in the present manuscript suggest that Cyclin B1 might itself be a Mad1 docking factor at kinetochores. Maybe the authors would like to comment/discuss on this possibility.

6- Discussion, model: the authors model clearly proposes that unattached kinetochores are the source of MCC and Cyclin B1-Mad1 interaction is critical to regulate the transition of Mad1 from NPCs to kinetochores in a Tpr-dependent manner. This should be discussed in the context of previous works by the Jalepalli and Maiato labs. The work of Lee et al, 2008 cited in the present work, should also be discussed as it supports an alternative model in which Tpr at kinetochores works as a docking factor for Mad1/Mad2.

Rebuttal: 201907082

We thank the reviewers for their thoughtful and constructive comments on our manuscript. We believe that we have addressed all the comments, either with new data or by clarifying the text, as detailed below. The most substantial changes are:

- 1) We have used super-resolution microscopy to demonstrate that Cyclin B1 co-localises at the nuclear membrane with TPR in wild type cells and that this is markedly reduced in the MAD1 mutant cells (new Figure 7).
- 2) We repeated our mitotic arrest assays without sirDNA using DIC alone (new Figures 3 and supplemental S3). We found that including sirDNA altered the response of the cells to taxol and nocodazole, which introduced differences between clones and between experimental repeats. We note that the Higgins lab has reported that sirDNA induces DNA damage (doi:10.1038/s41598-018-26307-6), which may be relevant to our observation. By removing sirDNA from our mitotic arrest assays we removed the apparent variability between the clones that had concerned both us and the referees.
- 3) We include data quantifying the amount of Cyclin B1 by immuno-blotting and show that there is no change in levels between the parental clones and the MAD1 mutants. We include these data as a figure for the referees.
- 4) In addition, we include data for the referees showing that there is no correlation between the level of MAD1 and the strength of the checkpoint in siRNA and rescue experiments. Since these experiments were performed in HeLa cells we prefer not to include them in the paper, but we can add them at the referees' request.

We hope that the reviewers will agree with us that these new experiments have substantially improved our manuscript, and we are grateful to the Editor and the reviewers for their constructive critiques.

Point by point:

Reviewer #1 (Comments to the Authors (Required)):

Summary:

Jackman and colleagues here investigate the role of a CyclinB1/Cdk1-Mad1 physical interaction in the regulation of spindle assembly checkpoint (SAC) signaling in cultured human cells. This is an area of intense interest, as Mad1, a key signaling component of kinetochore-mediated checkpoint signaling during mitosis, is localized to the nuclear pores during interphase and how Mad1 is released from the pores is an unresolved question. Recently, the Barr lab has shown that Mad1 recruits CyclinB1/Cdk1 to unattached kinetochores in human cells and that this recruitment is necessary for robust SAC signaling (Alfonso-Pérez et al., 2019 JCB). Extending upon these findings, Jackman and colleagues identify key residues mediating the Mad1/CyclinB1 interaction. They go on to perform a number of microscopy assays to argue that loss of this interaction (through CRISPR generated mutations) leads to defects in checkpoint signaling, and critically, inability of Mad1 to disassociate from nuclear pores during mitosis. Based on these data, the authors propose that CyclinB1/ Cdk1 activity coordinates Mad1 release from the nuclear pores prior to nuclear envelope breakdown to regulate the SAC and maintain genomic stability.

These observations, if reproducible and robust, would certainly be of great interest to the field as they move it closer to a mechanistic understanding of how Mad1 localization is regulated to mediate the SAC. However, there are experimental and textual concerns that must be addressed before reevaluating whether or not CyclinB1 interactions regulate Mad1 localization and checkpoint response in human cells.

We thank the reviewer for their supportive comment.

Major Concerns:

1) A key experimental concern is in the generation of clonal mutants. The authors use Mad1 mutant clones 7D2 and 8B12 throughout the manuscript, but it is troubling that the control parental line was not simultaneously single cell cloned. This brings up the possibility that many of the phenotypes observed in the mutant clones aren't necessarily the result of the E53/56K mutations but are rather due to the selection process of generating a single cell clone.

We thank the referee for this comment and should have clarified that the parental cell line was also single cell cloned. We now make this clear in the text.

A further concern is in the heterogeneity of the clonal phenotypes. The data in figure 3 clearly show that mutant 8B12 completes unperturbed mitosis faster than the parent cell line but is not affected by nocodazole or paclitaxel treatment. This is in contrast to mutant 7D2, which has the same mitotic time as the parent but is sensitive to paclitaxel. Likewise, according to figure panels 4D and 4E, these clones have different levels of CyclinB1. Is it possible that these different levels of CyclinB1 in the clones is what is driving the observed phenotypes? Are these the only 2 clones that survived genome editing? If they are the only surviving clones then the observed mitotic phenotypes are less meaningful to us. These concerns can be addressed experimentally by repeating the experiments entailed in Figures 3 and 4 using several clonal parental lines. .

We thank the referee for highlighting this. We were also concerned by the apparent heterogeneity of the response and discovered that this was due to including sirDNA in our assays. Once we removed this, the clones had an identical response to both nocodazole and taxol (new data in Figures 3 and Supplemental Data S3). We note that the Higgins lab has previously reported that sirDNA can induce DNA damage (doi:10.1038/s41598-018-26307-6), which may be related to our observation.

With regard to the level of Cyclin B1 in the clones, we have carefully measured this by immunoblotting and there is no difference between the parental clones and the MAD1 mutants (see Figure 1 for referees). The data for the original Figure 4 were collected on different days for the parental and the MAD1 mutants, and variation in the laser alignment led to the apparent differences in signal level. The data in the revised figures were collected at the same time for the parental and mutant clones using 8-well slides.

With regards to the number of clones: we screened ~750 clones, and identified 14 clones with MAD1 mutations, 2 of which were homozygous point mutations, the other 12 of which had point mutations in one allele and a null on the second allele.

2) Figure 2A and B make it appear as though there is less overall Mad1 in the mutant clones. Couldn't lower levels of Mad1, and not the mutations therein, explain the slower recruitment

of Mad2 to kinetochores seen in Figure 5? This should be addressed experimentally or textually.

We thank the referee for this comment but we do not think MAD1 levels explain the SAC defect. A number of previous studies on MAD1 (e.g. from the Sorger doi.org/10.1016/j.devcel.2004.06.006 and Nilsson labs [doi:10.1002/embr.201338101](https://doi.org/10.1002/embr.201338101)) showed that MAD1 had to be depleted to very low levels to observe an effect on the SAC, and we confirmed these studies when we started this study (about 15 years ago). At this time, we were using HeLa cells for our studies and used siRNA and rescue to investigate the effect of MAD1 mutants MAD1 α and MAD1 β . This analysis revealed that SAC strength – as assayed by duration of mitotic arrest and by the kinetics of Cyclin B1 degradation – correlated with the splice form of MAD1 but not with its level. We include these data in Figure 2 for the referees. Since these experiments were performed in HeLa rather than RPE1 cells we would prefer to not to include them in the manuscript, but we can include them if the referee prefers.

3) As mentioned in point 1 above, the phenotypes assayed in figure 3 are not consistent across clones. Due to this, the authors should tone down the claim that 'Mad1 binding to CCNB1 is required for genomic stability' or provide additional sufficient evidence.

We thank the referee for this comment as it allowed us to determine that the apparent inconsistency was due to including sirDNA in our assays. New Figures 3 and S3 performed without sirDNA show that the clones are consistent in their response.

4) Fusion of CyclinB1 to pores in mutant Mad1 background does not rescue SAC response to WT levels. This suggests that the main role for Cyclin B1 interaction with Mad1 is not at the pore but is rather perhaps at the kinetochore, as suggested by Alfonso-Pérez et al. This should be tested by fusing CyclinB1 to the kinetochore in both clones. Given the already mentioned phenotypic clonal heterogeneity, it is concerning that the current manuscript only presents the pore-fusion experiment in 1 clone.

We thank the referee for this suggestion. We tried fusing Cyclin B1 to the kinetochore through Mis12 but found that this dramatically perturbed mitosis; therefore, we tried fusing Cyclin B1 to TPR, but found that this also perturbed the cells and they mostly delocalised the TPR-Cyclin B1 from the nuclear membrane. Thus, we were unable to perform the experiment suggested by the referee. Nevertheless, we hope that the referee will agree that a two-fold increase in mitotic arrest when we recruit Cyclin B1 to Pom121 is significant. Moreover, as explained below, the timing of release of MAD1/2 from the nuclear envelope in our rescue experiment means that we should not expect complete reversion to wild type SAC response. We have amended the text to discuss this point and the possibility that Cyclin B1 recruitment to kinetochores after NEBD might also contribute to the SAC. The heterogeneity between the clones disappeared once we omitted sir DNA from our assays.

This experiment should also examine the localization of Mad1 to determine whether that is also partially restored. The authors conclude that the partial SAC rescue is due to restoring the release of Mad1 but never assay its localization.

We thank the reviewer for this suggestion. This is quite a laborious experiment to perform (finding cells with the right expression level of POM121 to recruit Cyclin B1 to the NPC that entered mitosis during filming) but in all the MAD1 mutant cells that recruited Cyclin B1 to

the NPC, MAD1/MAD2 was released from the NPC 2 to 4 minutes before NEBD, whereas MAD1/MAD2 were only released at NEBD in non-targeted cells. We note that this release is not as early as the 10 minutes seen in parental cells, which may help to explain why we did not restore the SAC to wild type strength.

5) Throughout the main text, the authors use the observed phenotypes to argue that CyclinB1-Cdk1 activity is important for releasing Mad1 from the NPC. There is no data to formally support this statement (e.g. no use of Cdk1 inhibitors that phenocopy the E53/56K mutants). Please revise these statements throughout the text or show that it is indeed Cdk1 activity.

We thank the referee for this comment and have addressed this in the text. We are sure that the referee is aware that Cdk inhibitors will inhibit Cdk1 throughout the cell and thus cannot phenocopy the MAD1 mutant where only a small pool of Cyclin B1 is displaced from its normal localisation.

Minor Points

1. A summary of MS data should be included in the manuscript in some form.

We now include these data in a table and have submitted all the results to the PRIDE database.

2. The title is an overstatement. While Cyclin B/Mad1 binding regulates Mad1 release from the NPC as well as genomic stability, there is no clear evidence that they are linked. The tethering experiment never addressed chromosomal stability so this title should be a better description of the findings.

We agree with the referee and have changed the title.

3. The Mad1/TPR colocalization in figure 6 should be quantified.

We thank the referee for this comment and have quantified the data in new Figure 6. This also prompted us to use super-resolution microscopy to measure the co-localisation between Cyclin B1 and TPR in wild type and MAD1 mutant clones and these data are presented in new Figure 7.

4. The introduction was a bit lengthy about issues that weren't necessarily related to the findings and could be shortened.

We thank the referee for this comment and have tried to reconcile this with the comments of referee 3 suggesting some additional citations.

Reviewer #2 (Comments to the Authors (Required)):

In this paper, the authors identify a Mad1-Cyclin B interaction and investigate the behavior of a Mad1 mutant that fails to bind to Cyclin B1. This mutant Mad1 still localizes to the nuclear envelope and kinetochores, but fails to recruit Cyclin B1 to kinetochores. In addition, this Mad1 mutant mislocalizes to chromosomes outside of kinetochores and appears to colocalize with TPR, a component of the nuclear pore complex (NPC). The mutant cells also display increased genome instability and a defect in maintaining the SAC, particularly when combined with an Mps1 inhibitor, which may be due to a delay in Mad2 recruitment to

kinetochores. The authors propose a model in which Mad1 binding to Cyclin B1-CDK1 promotes Mad1 release from NPC before NEBD such that there is a pool of Mad1/Mad2 poised to be recruited to kinetochores. Overall, this is an interesting model, but much more data is needed to support it. In addition, recent work from the Barr lab published in the JCB (Alfonso-Pérez et al. 2019) conducted very similar studies on the Mad1-Cyclin B interaction and its role in checkpoint control. Based on both of these, I cannot recommend publication of this paper in the JCB.

Major comments

1. The presence of the Mad1-Cyclin B1 interaction is well established by the work in this paper and the recent paper from the Barr lab (Alfonso-Perez et al. 2019). However, whether this interaction is required at kinetochores, the nuclear envelope, in the cytoplasm, or all of the above remains unclear. For this paper, the authors make a strong case that this interaction is required at the nuclear envelope, but I found this to be the least convincing part of the paper. First, the authors state that Cyclin B1-CDK1 activity at the nuclear envelope promotes Mad1 release from the NPC and allows proper recruitment to kinetochores. However, Cyclin B1 does not show strong enrichment at the nuclear envelope before NEBD (see Figure 7A). Mad2 has also been shown to directly bind TPR, but Mad2 release from NPC appears to be independent of Cyclin B1-CDK1. Second, the most compelling experiment for this point is the attempt to rescue this through the artificial tethering of Cyclin B1-CDK1 to the nuclear envelope through POM121. However, the data for this "rescue" in Figure 7B seems quite weak. There is only a minor increase in mitotic duration in this construct, and it certainly does not approach the levels that occur in the parental cells.

We thank the referee for these comments: they prompted us to undertake an analysis of Cyclin B1 co-localisation with TPR using SORA super-resolution microscopy in which we compared the behaviour of wild type and mutant clones. This showed that at the time when Cyclin B1 is imported into the nucleus in prophase, a sub-population colocalises with TPR at the nuclear membrane and this is not seen in cells with the MAD1 mutation. These data are in new Figure 7 and we believe have significantly strengthened our study.

With regard to the rescue experiment, we hope that the referee will agree with us that a two-fold increase in mitotic arrest is significant, and ours is the only study on Cyclin B1-MAD1 that has addressed the specificity of Cyclin B1 localisation in this way. Nevertheless, we agree with the referee that this is not a complete rescue and in the Discussion we address the possibility that Cyclin B1 might have a further role at kinetochores, as suggested by work from Adrian Saurin and Andrea Musacchio, in addition to the Barr lab.

2. The overlap with the recent work published by the Barr lab is a consideration, as many aspects of these papers are similar (the Barr lab identified a similar interaction domain and tested the role of this interaction in checkpoint function, etc). Ultimately, this is an editorial decision, but the data in this paper is not that different. Here, the authors have tried to emphasize different roles for this interaction and to develop a distinct model with function at the nuclear envelope, but as indicated above this is the least compelling aspect of the paper.

We respectfully disagree with the referee: the Barr lab used a 100 amino acid N-terminal deletion of MAD1 in their studies, a mutant that in addition to losing the Cyclin B1-binding site also lacks the nuclear localisation domain of Mad1 and is defective at binding the nuclear pore even when targeted to the nucleus with an exogenous nuclear localisation signal (our

observations). By contrast, we have refined the Cyclin B1 binding domain to the extent that we can effectively eliminate Cyclin B1 binding by mutating just two essential residues.

3. In the discussion, the authors state that "Cyclin B1-CDK1 coordinates with Mps1 kinase" to promote release of Mad1 from the NPC. Has Mps1 been shown to localize to the nuclear envelope? Alternatively, the effect of Mps1 inhibition may be independent of Mad1 release from the NPC. Instead, Mps1-mediated phosphorylation is simply required to promote Mad1 localization to kinetochores (as is the case in yeast or as was proposed in the Barr lab paper). In fact, a low dose of reversine alone leads to a reduction in Mad1 localization to kinetochores in the absence of the Mad1 localization around the chromosomes that is observed with the mutant.

We thank the referee for raising this possibility and we have expanded on the potential roles for Mps1 in the Discussion. We note that the case for Mps1 releasing MAD1 has just been strengthened by a paper recently published in *The Journal of Cell Biology* from the Conde lab showing that Mps1 phosphorylates TPR/Megator in prophase *Drosophila* cells to release MAD1 (Cuhna-Silva et al., *J Cell Biol* 219: e201906039); we now reference this paper in the text. We also reference a study that shows immunofluorescent staining of human Mps1 at the nuclear pore (Lui et al 2003).

4. The authors assess the timing of Mad2 localization and see a delay in recruitment to kinetochores. What about timing of Mad1 recruitment to kinetochores? Maybe the mutant Mad1 is less efficient at recruiting Mad2, which would also result in delayed Mad2 recruitment?

We thank the referee for raising this possibility but we have not seen any indication in our experiments that our MAD1 mutant is impaired in binding and recruiting MAD2.

Minor comments

1. The data in Fig 3B-D is not very convincing to established that the Mad1 mutants have a weakened SAC. Although the authors state that there statistical differences, the plots look very similar, just more spread out. What about trying higher levels of nocodazole and/or taxol? These should cause a more potent arrest, thus making it easier to assess whether some cells fail to maintain the SAC. However, in this case, the SAC is fully active which may be a problem if the effect of the mutation is mild.

We thank the referee for this comment, which prompted us to repeat our experiments but omitting sirDNA. These data show that the two clones have an identical impairment in SAC response to both taxol and nocodazole (new Figures 3 and S3). The Higgins lab has reported that sirDNA induces DNA damage, and this may be relevant to its perturbation of our assays.

2. Fig 1A - In the IP for Mad1, very little Cyclin B1 is pulled down when compared to the amount of Mad1 observed with IP for Cyclin B1. The authors should comment on this point and explain this discrepancy.

We thank the referee and are sure that they are aware of the problems in reciprocity when performing IP-westerns caused by antibody accessibility. We now comment on this in the text and, as suggested by referee 3, we include a reference to Schweizer et al, 2013 who identified Cyclin B1 in anti-MAD1 immunoprecipitates.

3. The ability to generate the Mad1 mutants at the endogenous locus is powerful and a strength of this paper. However, based on the way that these were generated, these cell lines have undergone multiple divisions in the absence of the wild type protein leading to an accumulation of defects. The different behavior of the clones is notable in this regard. In an ideal world, the authors would compare this is a conditional strategy to replace Mad1. In the absence of this, they should at least comment on this caveat and consideration.

We thank the referee for raising this point but we hope they will agree that this concern is diminished by our new data showing that the clones behave the same when sir DNA is omitted.

4. Based on previous proteome-wide studies (curated at phosphosite.org), Mad1 is phosphorylated at two sites that display a CDK-like consensus (residues 423 and 428). One possibility is that this interaction serves to recruit Cyclin B- Cdk1 to Mad1 to aid in its phosphorylation. It would be helpful to comment on this point, or possibly test the consequences of eliminating CDK sites in Mad1.

We thank the referee for raising this interesting point. We did analyse MAD1 phosphorylation in wild type and our mutant cells but disappointingly did not see any significant difference in phosphorylation at these sites.

5. Fig S1C - The authors state that Mad1 α and Mad1 β from cDNAs migrated at the same molecular masses as the two forms in cells. However, the blot is not very convincing. They could repeat this experiment to generate a better gel, or they could simply remove this data and this point as it is clearly established based on their mutant data. It should be enough to just say that Mad1 α is the only one that binds Cyclin B1.

We thank the referee for this comment but prefer to keep the data in the paper. The apparent difference in migration is due to the amount of protein running higher than MAD1 in the input extract that is not present in IPs.

6. Figure 1C and S1E have other Mad1 mutants that are not mentioned in the text, e.g. the 5AA substitution mutant that disrupts binding and nuclear envelope localization. The authors should write a few lines in the text describing the rationale and behavior of these mutants

We thank the referee for pointing this out and have expanded the text to explain the 5x alanine mutant.

7. Fig 1 - the authors should mention in legend that both Cyclin B1 and Mad1 were IP'ed.

Done.

8. Fig 1C - please indicate on the figure that the different Mad1 constructs are expressed from a tetON-inducible promoter.

Done

9. Fig 2A-B - Why is there a shift in the Mad1 band between input and after IP, especially

since this shift does not occur with Mad2? Also why are the two isoforms of Mad1 not detectable in the input?

The shift in MAD1 is due to amount of protein in extracts that runs above MAD1 in the input, which is not in the IP. This does not apply to MAD2, because cells contain many fewer proteins around the size of MAD2. This IP is from RPE1 cells, which have very low levels of MAD1 β compared to HeLa cells.

10. Fig 2C - although the strongest effect is on Cyclin B1 localization to kinetochores, the Mad2 signal is also reduced with more diffuse signal in the cytoplasm.

The referee is correct and we quantify this in Figure 5B.

11. Fig 4D-E - I am confused when the authors state that the Cyclin B1-Venus is degraded with similar kinetics in wild-type and mutant cells. This may be the case in the 7D2 mutant, but the slope is different in the 8B12 mutant.

We thank the referee for pointing this out and have repeated these experiments. These data now show very similar behaviour between the two clones (new Figure 4).

12. Fig 5A - do the cells displayed in the figure also express MIS12-RFP670? If yes, please indicate in the legend.

Done.

13. Fig 6 - It is nice to see colocalization between Mad1 and TPR around chromosomes, but if indeed the mutant remains bound to TPR, the authors should be able to observe that by immunoprecipitation.

We agree with the referee that this would be nice to show but IPs of MAD1 and TPR proved hard to interpret with our anti-MAD1 antibodies that do not efficiently co-immunoprecipitate TPR. We note that the TPR and MAD1 association has been previously shown by co-immunoprecipitation in Schweizer et al., 2013 and we now reference this in the text.

Reviewer #3 (Comments to the Authors (Required)):

The manuscript by Pines and colleagues shows how Cyclin B1 binding to Mad1 through an acidic patch on Mad1 regulates its dissociation from Tpr at the nuclear pore, required for efficient recruitment of Mad1/Mad2 to unattached kinetochores, and consequently normal SAC response, before complete nuclear envelope breakdown. The localization of Cyclin B1 to multiple mitotic structures, including the nuclear envelope and kinetochores has been known for years, but the relevance of its localization for function has remained unclear. More recently, work by the Barr lab has provided evidence for a functional interaction between Cyclin B1 and Mad1 required for SAC signaling. The present work by the Pines lab extends beyond current knowledge as it provides a view of how Cyclin B1-Mad1 interaction, by defining the precise interaction motif on Mad1, is required to mount a SAC response while inhibiting APC at mitotic entry. Importantly, the results provided in the current manuscript clarify previous controversy regarding the role of nuclear pores in generating an MCC (Rodriguez-Bravo et al., 2014). Overall, I find this an elegant study and an important

contribution that will be of interest to the JCB readership. I have only minor requests/suggestions that would like to see addressed prior to publication:

We thank the referee for their very supportive comments.

1- The introduction of this manuscript is an art piece as it easily, yet comprehensively, summarizes years of literature in a rather mature field, while clearly isolating the question being addressed: how disassembly of interphase structures are coordinated with the assembly of mitosis-specific structures, relevant for SAC signaling. I have only two suggestions: first, the authors refer to the role of Cyclin B-Cdk1 in the phosphorylation of structural components, including microtubule motors, but they do not mention non-motor MAPs, which probably account for the more dynamic behavior of microtubules as cells enter mitosis; second, in parallel to the work by the Jalepalli lab (Rodriguez-Bravo et al., 2014), it would be fair to cite also the previous works by the Maiato lab on Tpr and SAC regulation (Lince-Faria et al., 2009 and Schweizer et al., 2013), which offered an alternative model of how Tpr at nuclear pores regulates subsequent kinetochore-dependent MCC assembly and SAC response, in a way very consistent with the model being now proposed by the Pines lab. In fact, the original interaction between Mad1 and Cyclin B1-Cdk1 was reported (yet not functionally explored) by Schweizer et al., 2013, precisely using a similar mass-spec approach, but pulling from Mad1, which nicely complements the strategy used in this and previous works (Barr lab) in which Mad1 was identified after pulling from Cyclin B1.

We thank the referee for their very scholarly comments and now cite these studies as suggested.

2- The authors use low doses of nocodazole to infer SAC function in the Mad1 E53/56K mutant, but they reported no effect on SAC response. Arguably, low nocodazole is known to stabilize microtubules (M.A. Jordan, D. Thrower, L. Wilson Journal of Cell Science 1992 102: 401-416) and the real test to SAC response would be incubation of the WT vs mutant cells in high (micromolar) doses of nocodazole. This would also test about the issue of SAC maintenance.

We thank the referee for this suggestion but we have now repeated our assays in the absence of sirDNA and find that we do see a response in low doses of nocodazole. We are unsure why sirDNA perturbed our assays but the Higgins lab have previously reported that it can induce DNA damage.

3- Related to the previous point, the authors argue through elegant live-cell experiments that Mad2 recruitment was delayed in the Mad1 mutant cells and this was responsible for the compromised SAC response. An experiment that would further test this hypothesis would be to delay mitosis in the mutant by a temporary inhibition of the proteasome or APC activity, say for about 10 min, and then washout the inhibitor(s) and see if now Mad2 is recruited to normal levels and infer the respective SAC response and segregation fidelity.

We thank the referee for this suggestion. We agree that this would be an insightful experiment but our experience with washing out proteasome inhibitors and with the efficacy of APC/C inhibitors made this too much of a challenge to obtain sufficient cells for meaningful interpretation.

4- The authors refer to the role of Mps1 in Mad1 localization at unattached kinetochores only before Mad1 release from the NPC and cite prior works on Mps1 inhibition before or after mitotic entry (Hewitt et al., 2010). However, other works have shown that Mad1 recruitment and maintenance both depend on Mps1 activity (e.g. Schweizer et al., JCB 2013). The authors should therefore use caution here, as the relationship with Mps1 might not be so straightforward.

We thank the referee for this advice. We have modified the text and also reference a recent paper from the Conde lab showing that Mps1 releases MAD1 from TPR in *Drosophila* prophase.

5- The recruitment of Mad1 to kinetochores has been a matter of intense investigation recently. The results provided in the present manuscript suggest that Cyclin B1 might itself be a Mad1 docking factor at kinetochores. Maybe the authors would like to comment/discuss on this possibility.

We thank the referee for this suggestion and now discuss recent data from Adrian Saurin and Andrea Musacchio, who make exactly this observation.

6- Discussion, model: the authors model clearly proposes that unattached kinetochores are the source of MCC and Cyclin B1-Mad1 interaction is critical to regulate the transition of Mad1 from NPCs to kinetochores in a Tpr-dependent manner. This should be discussed in the context of previous works by the Jalepalli and Maiato labs. The work of Lee et al, 2008 cited in the present work, should also be discussed as it supports an alternative model in which Tpr at kinetochores works as a docking factor for Mad1/Mad2.

We thank the referee for this advice and now structure our Discussion accordingly.

Figure 1 for referees.

Figure 1 for referees.

RPE Cyclin B1-Venus+/-: Ruby-MAD2+/- Mad1 wt cells (Parental) and RPE Cyclin B1-Venus+/-: Ruby-MAD2+/- MAD1 E53K/E56K+/+ clones 7D2 and 8B12 have similar amounts of Cyclin B1 and Cyclin B1-Venus.

(A) Lysates from asynchronous cells were Western blotted and probed with anti-cyclin B1 (upper panel) and anti-hsp70 (lower panel) antibodies and assayed on a LiCOR Odyssey scanner.

(B) Quantification of summed cyclin B1 and cyclin B1-venus immunoblot bands from three independent experiments.

Figure 2 for referees.

Mad1 beta supports a less robust spindle assembly checkpoint compared to cells expressing similar amounts of Mad1 alpha.

Mad1 beta supports a less robust spindle assembly checkpoint compared to cells expressing similar amounts of Mad1 alpha.

Hela B1-YFP +/- cells stably transfected with either siRNAi-resistant Mad1 isoforms of mRuby2-Mad alpha or beta were transfected (twice, first day one and then day three with siRNAi oligos against Mad1 corresponding to the sequence (CAGCGATTGUGAAGAACAT, Ambion, ThermoFisher). Cells were synchronised with a double Thymidine block from the second siRNAi transfection on day three. At release from the second thymidine block on day five cells were treated with 60 nM Taxol (A and B) or 200nM Reversine with 100nM Nocodazole (C and D) and filmed from 8 hours later by confocal microscopy. NEBD-Anaphase was measured from CyclinB1-YFP localisation and stability. The relative amounts of mRuby2-Mad1 alpha or beta expressed in each cell was measured from mRuby2 fluorescence at the point of NEBD. Note, Mad1 -beta was N-terminally tagged with nuclear localisation signal (NLS), otherwise absent from this isoform. Exogenous NLS-Mad1 beta, in the absence of endogenous Mad1, localises to the nuclear envelope.

Legends for the three movies for referees:

Movies show projected z-series stacks of cells filmed using confocal spin disk microscopy at 2 minute time frames. Panels left to right show: Cyclin B1-YFP; Mad2-Ruby; RFP670-Mis12 and the merged image. Merged image: Mad2-Ruby (green), RFP670-Mis12 (red).

Movie 1; Wild type cells with Pom121-GBP

Movie 2: E53/56K mutant with Pom121-GBP

Movie 3: E53/56K mutant

February 27, 2020

RE: JCB Manuscript #201907082R-A

Prof. Jonathon Pines
Institute of Cancer Research
Cancer Biology
237 Fulham Road
London SW3 6JB
United Kingdom

Dear Prof. Pines:

Thank you for submitting your revised manuscript entitled "Cyclin B1-Cdk1 binds MAD1 and facilitate its release from the nuclear pore complex to ensure a robust Spindle Assembly Checkpoint". The paper has now been seen by the original reviewers again and they all recommend acceptance. Thus, we would be happy to publish your paper in JCB pending final revisions necessary to meet our formatting guidelines (see details below).

Please be sure to address the remaining (minor) comments of reviewer #3 (who, as you'll see, opted to sign his review).

A. MANUSCRIPT ORGANIZATION AND FORMATTING:

Full guidelines are available on our Instructions for Authors page, <http://jcb.rupress.org/submission-guidelines#revised>. **Submission of a paper that does not conform to JCB guidelines will delay the acceptance of your manuscript.**

1) Text limits: Character count for Articles and Tools is < 40,000, not including spaces. Count includes title page, abstract, introduction, results, discussion, and acknowledgments. Count does not include materials and methods, figure legends, references, tables, or supplemental legends. You are currently below this limit but please bear it in mind when revising.

2) Figure formatting: Scale bars must be present on all microscopy images, including inset magnifications. Molecular weight or nucleic acid size markers must be included on all gel electrophoresis.

3) Statistical analysis: Error bars on graphic representations of numerical data must be clearly described in the figure legend. The number of independent data points (n) represented in a graph must be indicated in the legend. Statistical methods should be explained in full in the materials and methods. For figures presenting pooled data the statistical measure should be defined in the figure legends. Please also be sure to indicate the statistical tests used in each of your experiments (both in the figure legend itself and in a separate methods section) as well as the parameters of the test (for example, if you ran a t-test, please indicate if it was one- or two-sided, etc.). Also, since you

used parametric tests in your study (e.g. t-tests, ANOVA, etc.), you should have first determined whether the data was normally distributed before selecting that test. In the stats section of the methods, please indicate how you tested for normality. If you did not test for normality, you must state something to the effect that "Data distribution was assumed to be normal but this was not formally tested."

4) Materials and methods: Should be comprehensive and not simply reference a previous publication for details on how an experiment was performed. Please provide full descriptions (at least in brief) in the text for readers who may not have access to referenced manuscripts. The text should not refer to methods "...as previously described."

5) Please be sure to provide the sequences for all of your primers/oligos and RNAi constructs in the materials and methods. You must also indicate in the methods the source, species, and catalog numbers (where appropriate) for all of your antibodies.

6) Microscope image acquisition: The following information must be provided about the acquisition and processing of images:

a. Make and model of microscope

b. Type, magnification, and numerical aperture of the objective lenses

c. Temperature

d. imaging medium

e. Fluorochromes

f. Camera make and model

g. Acquisition software

h. Any software used for image processing subsequent to data acquisition. Please include details and types of operations involved (e.g., type of deconvolution, 3D reconstitutions, surface or volume rendering, gamma adjustments, etc.).

7) References: There is no limit to the number of references cited in a manuscript. References should be cited parenthetically in the text by author and year of publication. Abbreviate the names of journals according to PubMed.

8) Supplemental materials: There are strict limits on the allowable amount of supplemental data. Articles/Tools may have up to 5 supplemental figures. At the moment, you meet this limit but please bear it in mind when revising.

Please also note that tables, like figures, should be provided as individual, editable files.

A summary of all supplemental material should appear at the end of the Materials and methods section.

9) eTOC summary: A ~40-50 word summary that describes the context and significance of the findings for a general readership should be included on the title page. The statement should be written in the present tense and refer to the work in the third person. It should begin with "First author name(s) et al..." to match our preferred style.

10) Conflict of interest statement: JCB requires inclusion of a statement in the acknowledgements regarding competing financial interests. If no competing financial interests exist, please include the following statement: "The authors declare no competing financial interests." If competing interests are declared, please follow your statement of these competing interests with the following statement: "The authors declare no further competing financial interests."

11) ORCID IDs: ORCID IDs are unique identifiers allowing researchers to create a record of their various scholarly contributions in a single place. At resubmission of your final files, please consider providing an ORCID ID for as many contributing authors as possible.

B. FINAL FILES:

-- High-resolution figure and video files: See our detailed guidelines for preparing your production-ready images, <http://jcb.rupress.org/fig-vid-guidelines>.

Thank you for this interesting contribution, we look forward to publishing your paper in Journal of Cell Biology.

Sincerely,

Arshad Desai, PhD
Monitoring Editor
Journal of Cell Biology

Tim Spencer, PhD
Executive Editor
Journal of Cell Biology

Reviewer #1 (Comments to the Authors (Required)):

The authors have done a good job addressing the bulk of the criticisms. I'd still prefer a focused intro that is not a review of the field but realize there are different opinions.

Reviewer #2 (Comments to the Authors (Required)):

This paper is much improved, and the authors have satisfactorily addressed my prior comments. The results section is much clearer now and more logical, and the experimental and textual additions and changes have made the paper much stronger. I also appreciated that they highlighted that their mutants were point mutations vs the deletions from the other related paper.

Reviewer #3 (Comments to the Authors (Required)):

The authors have addressed most of my previous concerns and I now recommend publication pending the clarification of few minor points:

1- The authors use cyclin-cdk, cyclin B-Cdk1, cyclin B1-Cdk1 interchangeably throughout the text. Please adopt a single, most accurate nomenclature.

2- The discussion of previous works on Tpr/Megator is not accurate. *Drosophila* Megator and human Tpr were NOT shown to localize to attached or unattached kinetochores by Lince-Faria et al., JCB, as opposed to the model proposed by Lee et al., G&D. As alluded in my original assessment, the model proposed by Lince-Faria et al, and later substantiated by Schweizer et al., JCB, proposes that (citing): "Tpr is a kinetochore-independent, rate-limiting factor required to mount and sustain a robust SAC response". In fact, it is clearly stated that (citing): "Our results are not consistent with a model in which KT-associated Tpr serves as a docking place for Mad1 because we were unable to detect Tpr (or Mtor) at KTs, including those that were positive for Mad1". This contrasts with the model proposed by Lee et al, and is fully consistent with the model proposed now by Pines and colleagues in the context of the Mad1-Cyclin B1 interaction. This should be clarified in the discussion.

Helder Maiato